# LARGE LANGUAGE MODEL EVALUATION VIA MATRIX NUCLEAR-NORM

## ABSTRACT

As large language models (LLMs) continue to evolve, efficient evaluation metrics are vital for assessing their ability to compress information and reduce redundancy. While traditional metrics like Matrix Entropy offer valuable insights, they are computationally intensive for large-scale models due to their $O(n^3)$ time complexity with Singular Value Decomposition (SVD). To mitigate this issue, we introduce the Matrix Nuclear-Norm, which not only serves as a metric to quantify the data compression proficiency of LLM but also provides a convex approximation of matrix rank to capture both predictive discriminability and diversity. By employing the $L_{1,2}$-norm to further approximate the nuclear norm, we can effectively assess the model's information compression capabilities. This approach reduces the time complexity to $O(n^2)$ and eliminates the need for SVD computation. Consequently, the Matrix Nuclear-Norm achieves speeds 8 to 24 times faster than Matrix Entropy for the CEREBRAS-GPT model as sizes increase from 111M to 6.7B. This performance gap becomes more pronounced with larger models, as validated in tests with other models like Pythia. Additionally, evaluations on benchmarks and model responses confirm that our proposed Matrix Nuclear-Norm is a reliable, scalable, and efficient tool for assessing LLMs' performance, striking a balance between accuracy and computational efficiency.

## 1 INTRODUCTION

Large language models (LLMs), such as Gemini (Gemini et al., 2023), Llama (Touvron et al., 2023), and GPT-4 (GPT-4 Achiam et al., 2023), have demonstrated remarkable performance across a variety of natural language processing (NLP) tasks (Zhao et al., 2023). They are not only transforming the landscape of NLP (Saul et al., 2005; Liu et al., 2023; Sawada et al., 2023) but also bring beneficial impacts on computer vision (Lian et al., 2023a; Wang et al., 2024) and graph neural networks (Zhang et al., 2024; Chen et al., 2024), achieving stellar results on various leaderboards. Despite these advancements, assessing a model's ability to compress information remains a crucial research challenge (Delétang et al., 2023). Compression focuses on efficiently distilling essential information from vast training datasets while discarding redundant elements, showcasing a model's ability to learn and recognize the underlying structure of the data (Wei et al., 2024). LLMs are expected to perform this form of compression during their training process (Zhao et al., 2023). Specifically, in the early stages of training, after random initialization, the representations produced from the data are often chaotic. However, as training progresses, these representations become more organized, allowing the model to filter out unnecessary information. Hence, assessing an LLM's capacity for information compression is crucial for understanding its learning efficiency and representational power.

Current compression evaluation methods, such as Matrix Entropy introduced by Wei et al. (2024), measure information compression efficiency through processing models' output representations on datasets. However, the reliance of Matrix Entropy on Singular Value Decomposition (SVD) (Kung et al., 1983; Zhang, 2015) leads to significant computational complexity, typically $O(n^3)$, which limits its applicability in large-scale models.

To tackle this challenge, we propose a novel evaluation metric called Matrix Nuclear-Norm. This metric effectively measures predictive discriminability and captures output diversity, serving as an upper bound for the Frobenius norm and providing a convex approximation of the matrix rank. Fur-

thermore, we enhance the Matrix Nuclear-Norm by employing the $L_{1,2}$-norm to approximate the nuclear norm, addressing stability issues during evaluation across multiple classes. This approach enables an efficient assessment of a model's compression capabilities and redundancy elimination abilities, streamlining the evaluation process. Notably, the Matrix Nuclear-Norm achieves a computational complexity of $O(n^2)$, a significant improvement over Matrix Entropy's $O(n^3)$. This reduction facilitates faster evaluations, making the Matrix Nuclear-Norm a practical choice for large-scale models while maintaining accuracy.

To validate the effectiveness of the Matrix Nuclear-Norm, we first conducted preliminary experiments on two language models of differing sizes. The results indicated a consistent decrease in Matrix Nuclear-Norm values as model size increased, signifying enhanced compression capabilities. Subsequently, we performed inference experiments on two widely used benchmark datasets, AlpacaEval (Dubois et al., 2024) and Chatbot Arena (Chiang et al., 2024), which cover a diverse range of language generation tasks. These benchmarks facilitate a comprehensive assessment of model inference performance. Our experimental findings confirm that the Matrix Nuclear-Norm accurately measures model compression capabilities and effectively ranks models based on performance, demonstrating its reliability and efficiency in practical applications. Our empirical investigations yield the following insights:

1. **Proposal of the Matrix Nuclear-Norm**: We present a new method that leverages the nuclear norm, successfully reducing the computational complexity associated with evaluating language models from $O(n^3)$ to $O(n^2)$. This reduction minimizes dependence on SVD, making the Matrix Nuclear-Norm a more efficient alternative to Matrix Entropy.

2. **Extensive Experimental Validation**: We validated the effectiveness of the Matrix Nuclear-Norm on language models of various sizes. Results indicate that this metric accurately assesses model compression capabilities, with values decreasing as model size increases, reflecting its robust evaluation capability.

3. **Benchmark Testing and Ranking**: We conducted inference tests on widely used benchmark datasets, AlpacaEval and Chatbot Arena, evaluating the inference performance of models across different sizes and ranking them based on the Matrix Nuclear-Norm. The results demonstrate that this metric can efficiently and accurately evaluate the inference performance of medium and small-scale models, highlighting its broad application potential in model performance assessment.

## 2  RELATED WORK

**LLM Evaluation and Scaling Laws.** Evaluating large language models (LLMs) is a multifaceted challenge, as it requires capturing both task-specific performance and internal representational efficiency. Scaling laws have become a foundational framework for studying how LLM performance evolves with model size and data volume (Kaplan et al., 2020; Ruan et al., 2024). These studies demonstrate that model performance on tasks like language modeling and fine-tuning often follows predictable power-law relationships with respect to model parameters and dataset size, emphasizing the importance of scaling for achieving state-of-the-art results.However, scaling laws typically focus on external metrics such as cross-entropy loss, offering limited insight into how LLMs manage internal knowledge representation. For instance, the ability of LLMs to compress knowledge, eliminate redundancy, and retain structured information remains poorly understood with traditional methods. Addressing these gaps requires structural metrics that go beyond task outcomes to directly evaluate the internal embeddings and activation patterns of LLMs.

**LLM Evaluation Metrics.** Traditional evaluation metrics such as perplexity, BLEU (Papineni et al., 2002), and ROUGE (Lin, 2004) primarily measure task-specific outcomes, assessing how well model outputs align with ground truth data. While these metrics are effective for evaluating surface-level outputs, they do not capture the underlying mechanisms of LLMs, such as the diversity or compression of embeddings. Similarly, accuracy and F1 score (Sasaki, 2007) focus on classification performance, making them less applicable to the generative tasks typical of LLMs.To bridge this gap, structural metrics such as Matrix Entropy have been introduced. Matrix Entropy (Wei et al., 2024) employs information theory to assess the entropy of covariance matrices derived from LLM embeddings. This metric evaluates how effectively a model removes redundancy and encodes structured information, offering a measure of its compression capabilities. For instance,

Matrix Entropy can reveal differences in embedding distributions across models of varying sizes, reflecting their capacity to extract meaningful patterns from large datasets. However, its reliance on Singular Value Decomposition (SVD) results in a computational complexity of $O(n^3)$, limiting its applicability to modern large-scale models. To overcome these limitations, we propose the Matrix Nuclear-Norm as a scalable alternative. By leveraging the $L_{1,2}$ norm as a convex approximation of matrix rank, the Matrix Nuclear-Norm reduces computational complexity to $O(n^2)$. This makes it feasible for evaluating embeddings from large-scale LLMs while preserving the insights provided by Matrix Entropy, such as compression efficiency.

## 3 PRELIMINARIES

This section presents the fundamental concepts used in our study to assess model performance, specifically focusing on discriminability, diversity, and the nuclear norm.

### 3.1 DISCRIMINABILITY MEASUREMENT: F-NORM

Higher discriminability corresponds to lower prediction uncertainty in the response matrix $A$, which can be quantified using Shannon Entropy (Shannon, 1948):

$$H(A) = -\frac{1}{B} \sum_{i=1}^{B} \sum_{j=1}^{C} A_{i,j} \log(A_{i,j}), \tag{1}$$

where $B$ represents the number of samples, and $C$ denotes the dimensionality of the output representation. The value $A_{i,j}$ represents the activation of the $j$-th dimension for the $i$-th sample. Minimizing $H(A)$ corresponds to maximizing discriminability, as lower entropy indicates less uncertainty in predictions. Alternatively, discriminability can be measured using the Frobenius norm of $A$, defined as:

$$\|A\|_F = \sqrt{\sum_{i=1}^{B} \sum_{j=1}^{C} |A_{i,j}|^2}. \tag{2}$$

The Frobenius norm reflects the overall magnitude of activations in $A$ and serves as a complementary metric to entropy. Higher $\|A\|_F$ implies stronger and more certain activations, indicating greater discriminability.

**Theorem 1.** For a matrix $A$ with non-negative elements, $H(A)$ and $\|A\|_F$ are strictly inversely monotonic. The proof is provided in Appendix A.5. Thus, minimizing $H(A)$ is equivalent to maximizing $\|A\|_F$. The bounds for $\|A\|_F$ are given as:

$$\sqrt{\frac{B}{C}} \leq \|A\|_F \leq \sqrt{B}, \tag{3}$$

where the lower bound corresponds to maximum uncertainty (e.g., uniform activation across all dimensions), and the upper bound corresponds to minimum uncertainty (e.g., one-hot activation).

This formulation ensures $H(A)$ and $\|A\|_F$ effectively evaluate LLMs in generation tasks, providing insights into discriminability and representation quality.

### 3.2 DIVERSITY MEASUREMENT: MATRIX RANK

In LLMs, diversity reflects the model's ability to utilize its latent representation space effectively, rather than predefined "categories" as in classification tasks. For a given dataset $\mathcal{D}$, the expected diversity of outputs, denoted as $E_C$, is defined as:

$$E_C = \mathbb{E}_{A \sim \mathcal{D}}(C_p(A)). \tag{4}$$

To approximate $C_p(A)$, we leverage the rank of $A$:

$$C_p(A) = \text{rank}(\prod[A_{i,\arg\max(A_i)}]) \approx \text{rank}(A). \tag{5}$$

Here, $\text{rank}(A)$ estimates the active subspace of the output embeddings. The maximum value of $C_p(A)$ is $\min(B, C)$, where $C$ is the dimensionality of the output representation. Maximizing $C_p(A)$ ensures effective utilization of the representation space, promoting robustness and reducing redundancy in generated outputs.

### 3.3 NUCLEAR NORM

The nuclear norm is an important measure related to diversity and discriminability.

**Theorem 2.** When $\|A\|_F \leq 1$, the convex envelope of $\text{rank}(A)$ is the nuclear norm $\|A\|_\star$. The theorem is proved in Fazel (2002).

The nuclear norm $\|A\|_\star$, defined as the sum of singular values of $A$, has significant implications for assessing model performance. With $\|A\|_F \leq \sqrt{B}$, we have:

$$\frac{1}{\sqrt{D}}\|A\|_\star \leq \|A\|_F \leq \|A\|_\star \leq \sqrt{D} \cdot \|A\|_F, \tag{6}$$

where $D = \min(B, C)$. Therefore, maximizing $\|A\|_\star$ ensures high diversity and discriminability.

The upper bound of $\|A\|_\star$ is given by:

$$\|A\|_\star \leq \sqrt{D \cdot B}. \tag{7}$$

## 4 METHODOLOGY

### 4.1 MOTIVATION

This section introduces the Matrix Nuclear-Norm, a novel metric designed to enhance the efficiency of model evaluation. Traditional nuclear norm calculations rely on computing all singular values, which typically involves the computationally intensive SVD. This method not only consumes significant time for large-scale data but may also fail to converge in certain cases, severely impacting practical application efficiency. Therefore, we propose the Matrix Nuclear-Norm, which utilizes the $L_{1,2}$-norm to approximate the nuclear norm, effectively eliminating computational bottlenecks. This innovation significantly reduces computational demands and ensures scalability, providing a robust framework for the LLM evaluation.

### 4.2 MATRIX NUCLEAR-NORM

Calculating the nuclear norm of a matrix $A \in \mathbb{R}^{B \times C}$ requires computing its Singular Value Decomposition (SVD), which has a time complexity of $O(\min(B^2C, BC^2))$, simplified to $O(n^3)$, where $n = \max(B, C)$. While manageable for smaller dimensions, this computation becomes infeasible for large-scale datasets and models. Moreover, SVD can fail to converge in certain cases, necessitating efficient approximations of singular values.

Since $A$ often exhibits sparsity in its activations, with significant values concentrated in a subset of dimensions, its singular values can be approximated by focusing on these dominant activations. This property enables efficient computation of metrics like the nuclear norm.

**Theorem 3.** When $\|A\|_F$ approaches its upper bound $\sqrt{B}$, the $j$-th largest singular value $\sigma_j$ can be approximated as:

$$\sigma_j \approx \text{top}\left(\sqrt{\sum_{i=1}^{B} A_{i,j}^2}, j\right), \quad \forall j \in \{1, \ldots, D\}. \tag{8}$$

The proof is detailed in Sect. A.6 of the Supplementary Materials. The batch nuclear norm can then be efficiently approximated as:

$$\|\hat{A}\|_* = \sum_{j=1}^{D} \text{top}\left(\sqrt{\sum_{i=1}^{B} A_{i,j}^2}, j\right).$$  (9)

Here, $\|\hat{A}\|_*$ represents the approximate nuclear norm of $\hat{A}$, with the "top" operation selecting the $D$ largest values from $\sqrt{\sum_{i=1}^{B} A_{i,j}^2}$. This approach effectively captures the dominant components while treating smaller contributions as noise.

This approach indicates that the primary components of the $L_{1,2}$-norm can effectively approximate the nuclear norm when $\|A\|_F$ is close to $\sqrt{B}$, while other components can be considered noise. Compared to traditional SVD-based methods (e.g., Guo et al. (2015)), this approach reduces computational complexity from $O(n^3)$ to $O(n^2)$ and avoids convergence issues by using only standard floating-point operations. The complete algorithm is detailed in Algorithm 1.

**Definition of Matrix Nuclear-Norm.** The approach can ultimately be expressed as:

$$\text{Matrix Nuclear-Norm}(\mathbf{X}) = \frac{\sum_{i=1}^{D}\left(\sqrt{\sum_{j=1}^{m} X_{i,j}^2}\right)}{L_{\text{input}}}$$  (10)

Here, $L_{\text{input}}$ denotes the length of the input sequence, ensuring comparability through normalization. Our observations indicate that Matrix Nuclear-Norm values increase with longer sequences; further details can be found in Section 5.3.2.

---

**Algorithm 1** Algorithm of Matrix Nuclear-Norm

---

**Require:** Sentence representations (hidden states from LLM) $\mathcal{S} = \{X_i\}_{i=1}^{m}$, where $X_i \in \mathbb{R}^{d \times 1}$, $d$ is the hidden dimension of representation, and $L_{\text{input}}$ is the length of the sentence.

1: $\mu = \frac{1}{m}\sum_{i=1}^{m} X_i$      // Calculate the mean embedding

2: $\mathbf{X}_{\text{norm}} = \frac{\mathbf{X}-\mu}{\|\mathbf{X}-\mu\|_{2,\text{row}}}$      // Normalize the activation matrix

3: $\text{L2}(\mathbf{X}_{\text{norm}}) = \sqrt{\sum_{i=1}^{m} \mathbf{X}_{i,j}^2}$      // Calculate $L_2$-norm for each column

4: $\Sigma_D = \{\sigma_1, \sigma_2, \ldots, \sigma_D\}$      // Sort $L_2$-norm and select top $D$

5: $\text{Matrix Nuclear-Norm}(\mathbf{X}) = \frac{\sum_{i=1}^{D}\left(\sqrt{\sum_{j=1}^{m} \mathbf{X}_{j,i}^2}\right)}{L_{\text{input}}}$      // Calculate Matrix Nuclear-Norm

6: **return** Matrix Nuclear-Norm

---

## 5 EXPERIMENTS OF LARGE LANGUAGE MODELS

The models and datasets used in this paper are thoroughly introduced in Appendix A.2.

### 5.1 BASELINES

**Cross-Entropy Loss.** Cross-entropy is a key metric for evaluating LLMs by measuring the divergence between predicted and true probability distributions. The formula is given as (Wei et al., 2024):

$$L = -\frac{1}{T}\sum_{i=1}^{T} \log P(u_i|u_{<i};\Theta)$$  (11)

where $u_i$ is the target word, $P(u_i|u_{<i};\Theta)$ is the predicted probability, and $T$ is the sequence length. Lower values indicate better prediction accuracy. We compare this baseline with the Matrix Nuclear Norm metric, using the same datasets and models from (Kaplan et al., 2020).

**Perplexity.** Perplexity measures how well a language model predicts a sequence of words. For a text sequence $U = \{u_1, \ldots, u_T\}$, it is defined as (Neubig, 2017; Wei et al., 2024):

$$PPL(U) = \exp\left(-\frac{1}{T}\sum_{i=1}^{T} \log P(u_i|u_{<i}; \Theta)\right) \tag{12}$$

Lower perplexity indicates better performance, showing that fewer attempts are needed to predict the next word.

**Matrix Entropy of a Dataset.** For a dataset $\mathcal{D} = \{\mathcal{S}_i\}_{i=1}^{n}$, where $\mathcal{S}_i$ represents sentence embeddings, the matrix entropy is defined as(Wei et al., 2024):

$$H(\mathcal{D}) = \frac{\sum_{i=1}^{n} H\left(\Sigma_{\mathcal{S}_i}\right)}{n \log d}, \tag{13}$$

where $\Sigma_{\mathcal{S}_i} = \sum_{j=1}^{d} \mathcal{S}_{i,j}$ is the sum of elements in the embedding $\mathcal{S}_i$, and $d$ is the embedding dimension. The normalization ensures the entropy reflects the diversity of embeddings in $\mathcal{D}$.

## 5.2 MATRIX NUCLEAR-NORM OBSERVATION

### 5.2.1 A COMPARATIVE ANALYSIS OF COMPUTATIONAL TIME

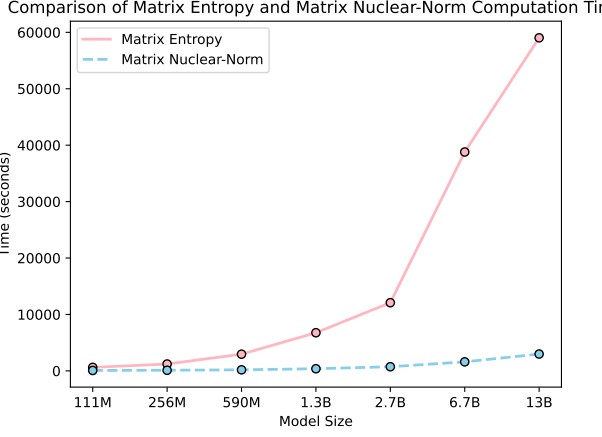

Figure 1: CEREBRAS-GPT: Time comparison

To evaluate the computational efficiency of Matrix Nuclear-Norm in comparison to Matrix Entropy for LLMs, we conducted experiments across various model sizes using multiple benchmark datasets. The results, summarized in Table 1, demonstrate a clear advantage of Matrix Nuclear-Norm in terms of computation time, particularly for larger models.

As model sizes increased, Matrix Entropy's computation time rose dramatically, reaching approximately 16.3 hours for the 13B model . In contrast, Matrix Nuclear-Norm only required about 0.82 hours for the same model, representing nearly a 20-fold reduction in computation time. This trend was consistent across all model sizes, with Matrix Nuclear-Norm consistently proving to be much faster (as illustrated in Figure 1). For example, the 111M model showed that Matrix Nuclear-Norm was 8.58 times quicker than Matrix Entropy.

The significant efficiency gain is due to the lower complexity of Matrix Nuclear-Norm, $O(m \cdot n + n \log n)$, versus Matrix Entropy's $O(n^3)$, where $m$ is the embedding dimension (columns). This makes it an efficient metric for LLM evaluation, especially for large-scale models.

In summary, Matrix Nuclear-Norm achieves comparable evaluation accuracy to Matrix Entropy but with vastly superior computational efficiency, making it a practical and scalable choice for assessing LLMs.

| Model Size | Matrix Entropy Time (s) | Matrix Nuclear-Norm Time (s) | Ratio |
|---|---|---|---|
| 111M | 623.5367 | 72.6734 | 8.5800 |
| 256M | 1213.0604 | 110.8692 | 10.9414 |
| 590M | 2959.6949 | 184.7785 | 16.0175 |
| 1.3B | 6760.1893 | 379.0093 | 17.8365 |
| 2.7B | 12083.7105 | 732.6385 | 16.4934 |
| 6.7B | 38791.2035 | 1598.4151 | 24.2685 |
| 13B | 59028.4483 | 2984.1529 | 19.7806 |

Table 1: CEREBRAS-GPT: Time Comparison between Matrix Entropy and Matrix Nuclear-Norm

### 5.2.2 SCALING LAW OF MATRIX NUCLEAR-NORM

To affirm Matrix Nuclear-Norm's efficacy as an evaluative metric, we evaluated Cerebras-GPT models on four datasets including dolly-15k, Wikipedia, openwebtext2, and hh-rlhf comparing Matrix Nuclear-Norm, matrix entropy, perplexity, and loss. Results, detailed in Table 10 (Appendix), demonstrate Matrix Nuclear-Norm's consistent decrease with model size enlargement, signifying better data compression and information processing in larger models. This trend (see in Figure 2b) validates Matrix Nuclear-Norm's utility across the evaluated datasets. Notably, anomalies at the 2.7B and 13B highlight areas needing further exploration.

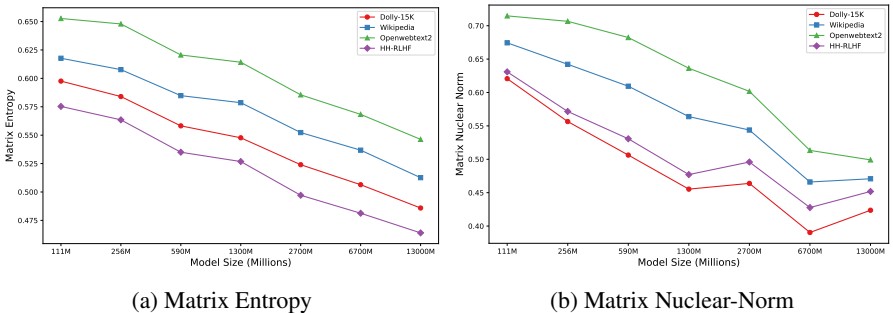

(a) Matrix Entropy          (b) Matrix Nuclear-Norm

Figure 2: Comparison of Matrix Nuclear-Norm, matrix entropy when model scales up.

### 5.2.3 RELATIONSHIP OF BENCHMARK INDICATORS

Findings indicate the efficacy of the Matrix Nuclear-Norm as a metric for evaluating LLM, as shown in Table 9 (Appendix), there is an overall downward trend in Matrix Nuclear-Norm values with increasing model sizes, signifying enhanced compression efficiency. However, notable anomalies at the 2.7B and 13B checkpoints suggest that these specific model sizes warrant closer examination. Despite these discrepancies, the Matrix Nuclear-Norm consistently demonstrates superior computational efficiency and accuracy compared to traditional metrics, highlighting its promising applicability for future model evaluations.

## 5.3 LANGUAGE INVESTIGATION

### 5.3.1 SENTENCE OPERATION EXPERIMENTS

Figure 3 clearly indicates that sentence manipulations significantly influence Matrix Nuclear-Norm values, which generally decline as model size increases. This trend confirms the enhanced information compression capabilities of larger models. The ranking of Matrix Nuclear-Norm values by operation is as follows: Reverse > Shuffle & Reverse > Shuffle > Base. This indicates that disrupting sentence structure through Reverse and Shuffle & Reverse operations leads to higher Matrix

Nuclear-Norm values due to increased information chaos and processing complexity. In contrast, the Shuffle operation has minimal effect on compression, while the Base condition consistently yields the lowest Matrix Nuclear-Norm values, signifying optimal information compression efficiency with unaltered sentences.

Despite the overall downward trend in Matrix Nuclear-Norm values with increasing model size, the 2.7B model exhibits slightly higher values for Shuffle and Base operations compared to the 1.3B model. This anomaly suggests that the 2.7B model may retain more nuanced information when handling shuffled data or operate through more intricate mechanisms. However, this does not detract from the overarching conclusion that larger models excel at compressing information, thereby demonstrating superior processing capabilities.

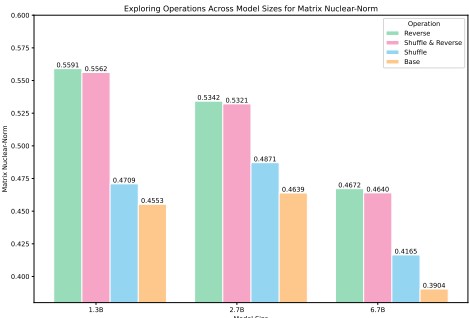

Figure 3: Results of sentence operation. Shuffling and reversing disrupt the text structure and diminish the informational content, leading to an increase in Matrix Nuclear-Norm.

### 5.3.2 ANALYSIS OF LENGTH DYNAMICS

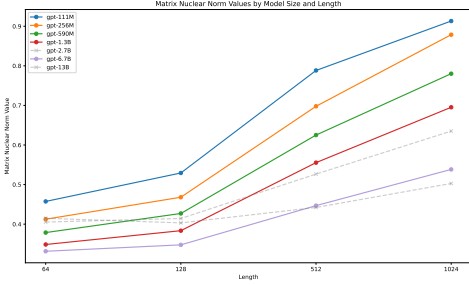

Figure 4: The Matrix Nuclear-Norm values increase consistently with longer text input lengths, reflecting the model's ability to capture more information.

The analysis reveals that Matrix Nuclear-Norm values generally increase as input length rises, aligning with our expectations (see Figure 4). Longer inputs necessitate that the model manage and compress more information, which naturally leads to higher Matrix Nuclear-Norm values. Most models exhibit this trend, indicating effective handling of the increased information load.

However, the gpt-2.7B and gpt-13B models display anomalies in their Matrix Nuclear-Norm values at 64 and 128 tokens, where the value at 128 tokens is lower than that at 64 tokens. This discrepancy may be attributed to these models employing different information compression mechanisms or optimization strategies tailored to specific input lengths, allowing for more effective compression at those lengths.

Overall, aside from a few outliers, the results largely conform to expectations, demonstrating that Matrix Nuclear-Norm values increase with input length, reflecting the greater volume and complexity of information that models must handle.To address the observed trend of rising Matrix Nuclear-Norm values with longer sentences, we incorporated a normalization step in our methodology via dividing the Matrix Nuclear-Norm values by the sentence length. This adjustment helps mitigate any biases introduced by models that tend to generate longer sentences during inference.

### 5.3.3 Analysis of Prompt Learning

The experimental results (shown in Table 2) indicate that we performed inference on different sizes of GPT models using three carefully selected prompts (shown in Table 12) and calculated the Matrix Nuclear-Norm values of their responses. As the model size increased, the Matrix Nuclear-Norm values gradually decreased, demonstrating that larger models possess greater information compression capabilities. The prompts significantly influenced Matrix Nuclear-Norm, with variations reflecting the models' responses to prompt complexity. Specifically, GPT-1.3B showed a notable decrease in Matrix Nuclear-Norm after the input prompts, indicating its sensitivity to them, while GPT-2.7B exhibited smaller changes. In contrast, GPT-6.7B displayed minimal variation across all prompts, suggesting stable performance regardless of prompt detail. Overall, more detailed prompts resulted in larger information volumes in the model's responses, leading to corresponding changes in Matrix Nuclear-Norm values.

Table 2: Results of prompt learning with (Empty Prompt) and without (Prompt 1, 2, 3) the use of prompts. Incorporating prompts as prefixes before the QA pairs enhances the models' ability to achieve better compression.

| MODELS | ADDING PROMPT TO QA PAIRS | | | | | |
|---|---|---|---|---|---|---|
| | EMPTY PROMPT | PROMPT 1 | PROMPT 2 | PROMPT 3 | AVERAGE | $\Delta x$ |
| CEREBRAS-GPT-1.3B | 0.150955 | 0.147577 | 0.140511 | 0.141358 | 0.14453 | ↓**0.006425** |
| CEREBRAS-GPT-2.7B | 0.150130 | 0.151522 | 0.142834 | 0.151842 | 0.14844 | ↓**0.001690** |
| CEREBRAS-GPT-6.7B | 0.132042 | 0.128346 | 0.124094 | 0.133211 | 0.12923 | ↓**0.002812** |

## 6 Implementing Proposed Metrics: Evaluating and Ranking Language Models in Practice

### 6.1 Inference-Based Model Assessment

In this section, we evaluated model inference across the AlpacaEval and Chatbot Arena benchmarks using the Matrix Nuclear-Norm metric prior to the final MLP classification head. The analysis revealed that Matrix Nuclear-Norm reliably ranks model performance, with lower values indicating enhanced information processing efficiency, particularly as model size scales up.

For instance, the Llama-3 70B model demonstrated superior compression capabilities compared to its 8B counterpart, as reflected by significantly lower Matrix Nuclear-Norm values across both benchmarks (see Table 8 in the Appendix). A similar trend was observed in the Vicuna family, where Matrix Nuclear-Norm values consistently decreased from 0.4623 for the 7B model to 0.3643 for the 33B model on the AlpacaEval dataset, indicating progressive improvements in information handling (see Table 3). Additionally, the DeepSeek models exhibited a consistent decrease in Matrix Nuclear-Norm values as model size increased, further demonstrating the metric's validity.

Overall, these results substantiate Matrix Nuclear-Norm as a robust and reliable tool for evaluating and ranking LLMs, demonstrating its capacity to capture critical aspects of model performance across diverse benchmarks.

| Model | DataSet | 7B | 13B | 33B |
|---|---|---|---|---|
| Vicuna | Alpaca | 0.4623 | 0.4159 | 0.3643 |
| | Arena | 0.4824 | 0.4311 | 0.3734 |

| Model | 1.3B | 6.7B | 7B |
|---|---|---|---|
| DeepSeek | 0.4882 | 0.3472 | 0.3352 |
| | 0.5754 | 0.4175 | 0.4357 |

Table 3: Matrix Nuclear-Norms in Vicuna and DeepSeek Responses

### 6.2 Matrix Nuclear-Norm Benchmarking: Ranking Mid-Sized Models

In this experimental section, we utilized Matrix Nuclear-Norm to evaluate the responses of LLMs, focusing on 7B and 70B variants. Notably, lower Matrix Nuclear-Norm values indicate more efficient information compression, serving as a robust indicator of model performance.

Among the 7B models, DeepSeek-7B exhibited the most efficient information processing with the lowest average Matrix Nuclear-Norm score of 0.3855 across Alpaca and Arena datasets (see Table

3). Gemma-7B followed closely with an average score of 0.3879, whereas QWEN 2-7B demonstrated less efficient compression with an average score of 0.5870. In contrast, the 70B models showed varied performance, with Llama 2-70B achieving the best average score of 0.3974, slightly outperforming Llama 3-70B (0.4951) and QWEN models, which scored around 0.5.

Interestingly, certain 7B models, like DeepSeek-7B and Gemma-7B, outperformed larger 70B models, underscoring that model efficiency is not solely determined by size. These results highlight that factors such as architecture, training methodology, and data complexity play crucial roles in information processing capabilities beyond scale.

| MODEL | Matrix Nuclear-Norm | | | Rank |
|---|---|---|---|---|
| | Alpaca | Arena-Hard | Avg Score | |
| DeepSeek-7B | 0.3352 | 0.4357 | 0.3855 | ↓ |
| Gemma-7B | 0.3759 | 0.3998 | 0.3879 | ↓ |
| Vicuna-7B | 0.4623 | 0.4824 | 0.4724 | ↓ |
| LLaMA 2-7B | 0.4648 | 0.5038 | 0.4843 | ↓ |
| QWEN 1.5-7B | 0.4866 | 0.5165 | 0.5016 | ↓ |
| Mistral-7B | 0.4980 | 0.5126 | 0.5053 | ↓ |
| QWEN 2-7B | 0.5989 | 0.5751 | 0.5870 | ↓ |
| QWEN 1.5-72B | 0.5291 | 0.5065 | 0.5178 | ↓ |
| QWEN 2-72B | 0.5261 | 0.4689 | 0.4975 | ↓ |
| Llama 3-70B | 0.4935 | 0.4967 | 0.4951 | ↓ |
| Llama 2-70B | 0.3862 | 0.4086 | 0.3974 | ↓ |

Table 4: Matrix Nuclear-Norm Rankings: A Comparative Analysis of Model Performance

To validate the design rationale and robustness of the Matrix Nuclear-Norm, we conducted a series of ablation studies. Due to space constraints, detailed results are provided in A.1 (appendix) to maintain brevity in the main text. These experiments included evaluations across different model families, such as Cerebras-GPT and Pythia, as well as comparisons of various data sampling strategies. The results demonstrate that the Matrix Nuclear-Norm consistently performs well across different model scales and sampling variations. This not only confirms its applicability across diverse models but also verifies its stability and reliability in handling large-scale datasets. We also provide an ablation study in the appendix, further proving the method's efficiency and accuracy in evaluating LLMs.

## 7 CONCLUSION

In conclusion, Matrix Nuclear-Norm stands out as a promising evaluation metric for LLMs, offering significant advantages in assessing information compression and redundancy elimination. Its key strengths include remarkable computational efficiency, greatly exceeding that of existing metrics like matrix entropy, along with exceptional stability across diverse datasets. Matrix Nuclear-Norm's responsiveness to model performance under varying inputs emphasizes its ability to gauge not only performance but also the intricate adaptability of models. This metric marks a significant advancement in NLP, establishing a clear and effective framework for future research and development in the evaluation and optimization of language models.

## 8 LIMITATIONS

Although Matrix Nuclear-Norm performs well in evaluating the performance of LLMs, it still has some limitations. First, since Matrix Nuclear-Norm's computation relies on the model's hidden states, the evaluation results are sensitive to both the model architecture and the training process. As a result, under different model designs or training settings, especially for models like GPT-1.3B and GPT-2.7B, inconsistencies in Matrix Nuclear-Norm's performance may arise, limiting its applicability across a wider range of models. Additionally, while Matrix Nuclear-Norm offers computational efficiency advantages over traditional methods, it may still face challenges with resource consumption when evaluating extremely large models. As model sizes continue to grow, further optimization of Matrix Nuclear-Norm's computational efficiency and evaluation stability is required.

## 9 ETHICS STATEMENT

Our study adheres to strict ethical guidelines by utilizing only publicly available and open-source datasets. We ensured that all datasets used, such as dolly-15k, hh-rlhf, OpenBookQA, Winogrande, PIQA, AlpacaEval, and Chatbot Arena, are free from harmful, biased, or sensitive content. Additionally, careful curation was conducted to avoid toxic, inappropriate, or ethically problematic data, thereby ensuring the integrity and safety of our research. This commitment reflects our dedication to responsible AI research and the broader implications of using such data in language model development.

## 10 REPRODUCIBILITY

We emphasize the importance of reproducibility in the development and evaluation of our newly proposed metric, Matrix Nuclear-Norm. To facilitate reproducibility, we provide detailed information regarding our data processing and parameter settings:
**Data Processing and Parameter Settings:** We outline the preprocessing steps applied to each dataset, ensuring that other researchers can accurately replicate our methodology. All hyperparameters and configuration settings used during the experiments are specified in the code, offering clarity on the experimental conditions.
**Experimental Procedures:** We detail the specific steps required to evaluate the Matrix Nuclear-Norm, including its application to each dataset and the metrics used for performance assessment.
**Code Availability:** Our implementation code, evaluation scripts, and pretrained models will be made publicly available upon acceptance of this paper, enabling others to reproduce our experiments and validate our findings.
By adhering to these guidelines, we aim to ensure that our work is accessible and reproducible for future research endeavors.

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

# A APPENDIX

## A.1 ABLATION STUDY

To thoroughly validate the rationale behind our metric design, experimental framework, and the efficacy of Matrix Nuclear-Norm, we conducted a series of ablation studies.

### A.1.1 DIFFERENT MODEL FAMILY

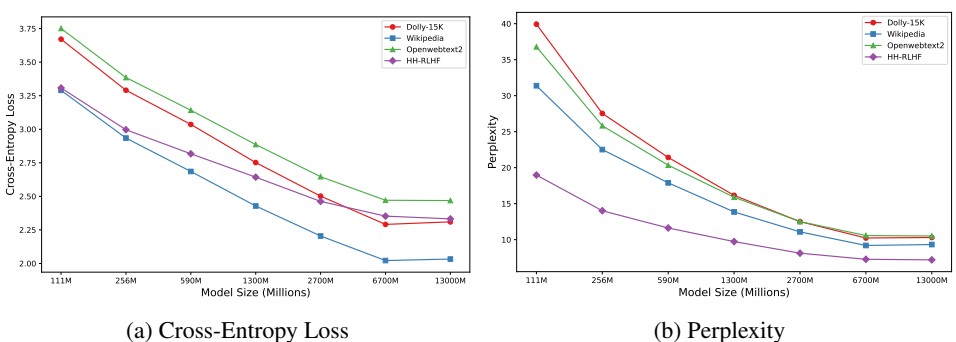

(a) Cross-Entropy Loss  (b) Perplexity

Figure 5: Comparison of loss, and perplexity when model scales up.

In addition to evaluating Matrix Nuclear-Norm within the Cerebras-GPT model series, we extended our experiments to the Pythia model family, which spans from 14M to 12B parameters and is trained on consistent public datasets. Utilizing the same datasets as described in Section 5.2.2, we computed matrix entropy, loss values, and Matrix Nuclear-Norm for these models. The empirical results (see Figure 6c) demonstrate that the Matrix Nuclear-Norm values for the Pythia models adhere to established scaling laws. However, we excluded metrics for the 14M, 31M, and 1B models due to notable deviations from the expected range, likely stemming from the inherent instability associated with smaller parameter sizes when tackling complex tasks. This further reinforces Matrix Nuclear-Norm as a robust metric for assessing model performance, underscoring its utility in the comparative analysis of LLMs.

Moreover, we compared the computation times for Matrix Entropy and Matrix Nuclear-Norm across the Pythia models (can see in Figure 6). The results unequivocally indicate that Matrix Nuclear-Norm necessitates considerably less computation time than Matrix Entropy, underscoring its efficiency. Detailed results are summarized in Table 11.

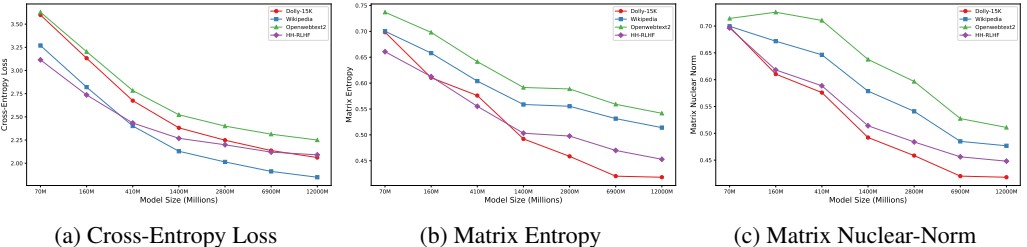

(a) Cross-Entropy Loss  (b) Matrix Entropy  (c) Matrix Nuclear-Norm

Figure 6: Pythia Model Metrics: Matrix Nuclear-Norm, Matrix Entropy, and Loss

### A.1.2 SAMPLING STRATEGY

In the ablation experiments, we extracted a baseline subset of 10,000 entries from the extensive Wikipedia dataset using three random seeds to evaluate the robustness of the Matrix Nuclear-Norm metric. We also tested additional subsets of 15,000 and 20,000 entries due to potential entry count issues. Given the large scale of the datasets, comprehensive calculations were impractical, so we employed random sampling.

The results showed that variations in random seeds and sample sizes had minimal impact on Matrix Nuclear-Norm values, with a standard deviation of only 0.0004975 (see Table 5), indicating high

consistency across trials. These findings confirm the Matrix Nuclear-Norm as a reliable metric for large-scale datasets, effectively evaluating information compression and redundancy elimination in LLMs.

Table 5: Ablation study of differnet sampling strategies on the Wikimedia(Foundation, 2024) dataset.

| MODEL | SAMPLING STRATEGY | | | | | STANDARD DEVIATION |
|---|---|---|---|---|---|---|
| | 10000 (SEED 1) | 10000 (SEED 2) | 10000 (SEED 3) | 15000 | 20000 | |
| CEREBRAS-GPT-1.3B | 0.5684 | 0.5670 | 0.5676 | 0.5699 | 0.5693 | 0.0004975 |

## A.2 MODEL SELECTION AND DATASETS FOR ANALYSIS

**Model Selection.** To investigate language model scaling, we employed a diverse set of transformer-based large language models (LLMs) across varying parameter sizes. A key focus of our analysis was the Cerebras-GPT model (Gao et al., 2020), which ranges from 111 million to 13 billion parameters, providing a comprehensive look at scaling effects in pre-trained models. Additionally, we included scaled versions of the Pythia model (Biderman et al., 2023), with parameter counts ranging from 14 million to 12 billion, enabling a broader analysis of model performance across different scales.

To ensure a well-rounded evaluation, we also tested a variety of models, including the DeepSeek series (1.3B, 6.7B, 7B) (Guo et al., 2024), Llama3 series (8B, 70B) (Dubey et al., 2024), QWEN 2 series (0.5B, 1.5B, 7B, 72B) (Yang et al., 2024), and Vicuna models (7B, 13B, 33B) (Chiang et al., 2023). For additional comparative insights, we included models of similar scale, such as Gemma-7B (Team et al., 2024) and Mistral-7B (Jiang et al., 2023).

**Datasets for Analysis.** Our experiments were conducted using several key benchmark datasets. We selected AlpacaEval(Dubois et al., 2024) and ChatBot Arena (Zheng et al., 2023) as the primary datasets for model evaluation. Additionally, subsets from Wikipedia (Foundation, 2024) and OpenWebText2 (Skylion007, 2019) were utilized to track variations in Matrix Nuclear-Norm values, especially with the Cerebras-GPT models.

To validate the Matrix Nuclear-Norm metric, we employed the dolly-15k dataset (Conover et al., 2023) for instruction tuning and the hh-rlhf dataset (Bai et al., 2022) for reinforcement learning with human feedback (RLHF). Further evaluations were performed on benchmark datasets such as OpenBookQA (Mihaylov et al., 2018), Winogrande (Sakaguchi et al., 2021), and PIQA (Bisk et al., 2020). Lastly, prompt learning experiments with the OpenOrca dataset (Lian et al., 2023b) provided a comprehensive framework for assessing the Matrix Nuclear-Norm's effectiveness across a variety of inference tasks.

## A.3 SUPPLEMENTARY EXPERIMENT RESULTS

The following results provide additional insights into the Matrix Nuclear-Norm evaluations and comparisons across various language models:

1. Tables 8 and 7 present the Matrix Nuclear-Norm evaluation results during the inference process for Llama-3 and QWEN-2.

2. Figure 7 illustrates that as model size increases, the computation time for Matrix Entropy grows exponentially, while Matrix Nuclear-Norm demonstrates a significant time advantage. This further emphasizes Matrix Nuclear-Norm's efficiency in assessing model performance.The complete results are presented in Table 6, which includes all relevant time data for the Pythia model family.

3. Table 10 contains the complete results for the comparison of Matrix Nuclear-Norm and other metrics based on Cerebras-GPT family considered in Figure 2b.

4. Table 9 demonstrates the correlation between Matrix Nuclear-Norm and other benchmark indicators, showing a consistent trend where values decrease as model size increases. This analysis examines the performance of language modeling indicators across OpenBookQA, Winogrande, and PIQA datasets.

5. Table 11 illustrates the numerical results of Figure 6c in the ablation study of Pythia family.

6. Table 12 shows the prompts used for the investigation of prompt learning.

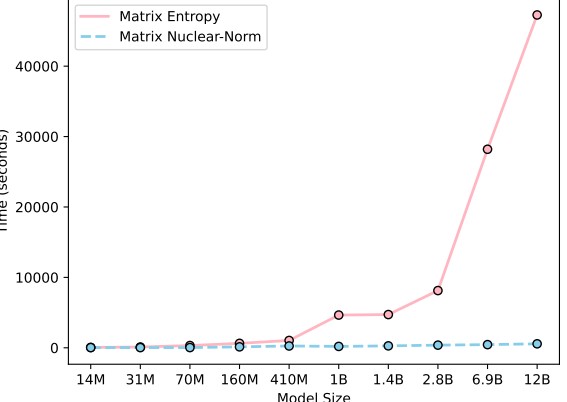

Figure 7: Pythia: Time Comparison of Matrix Entropy and Nuclear-Norm

| Model Size | Matrix Entropy Time (s) | Matrix Nuclear-Norm Time (s) | Ratio |
|---|---|---|---|
| 14M | 52.8669 | 22.2652 | 2.3772 |
| 31M | 114.0820 | 28.1842 | 4.0477 |
| 70M | 320.6641 | 24.3188 | 13.1855 |
| 160M | 631.9762 | 41.6187 | 15.1817 |
| 410M | 1040.9764 | 80.9814 | 12.8481 |
| 1B | 4650.1264 | 114.0639 | 40.8387 |
| 1.4B | 6387.0392 | 347.8670 | 18.3858 |
| 2.8B | 8127.1343 | 343.3888 | 23.6778 |
| 6.9B | 28197.8172 | 816.6332 | 34.5350 |
| 12B | 47273.5235 | 1276.1128 | 37.0485 |

Table 6: Pythia Model: Matrix Entropy vs. Matrix Nuclear-Norm Time Comparison

| Model | DataSet | 0.5B | 1.5B | 7B | 72B |
|---|---|---|---|---|---|
| QWEN 2 | Alpaca | 0.6551 | 0.6176 | 0.5989 | 0.5261 |
| | Arena | 0.6872 | 0.6374 | 0.5751 | 0.4689 |

Table 7: Matrix Nuclear-Norm in QWEN 2 Responses

| Model | 8B | 70B |
|---|---|---|
| Llama-3 | 0.5782 | 0.4935 |
| | 0.5817 | 0.4967 |

Table 8: Matrix Nuclear-Norm in Llama3 esponses

## A.4 ANALYSIS OF ALGORITHMIC COMPLEXITY

The primary computational expense of Matrix Nuclear-Norm arises from the calculation and sorting of the L2 norm of the matrix. By avoiding Singular Value Decomposition (SVD), we reduce the time complexity from the traditional nuclear norm of $O(n^3)$ to $O(n^2)$, giving Matrix Nuclear-Norm a significant advantage in handling large-scale data. This reduction in complexity greatly enhances the algorithm's practicality, especially for applications involving large matrices.

When analyzing the time complexity of the newly proposed Matrix Nuclear-Norm (L2-Norm Based Approximation of Nuclear Norm) against traditional Matrix Entropy, our objective is to demonstrate that Matrix Nuclear-Norm significantly outperforms Matrix Entropy in terms of time efficiency. We will support this claim with detailed complexity analysis and experimental results.

### A.4.1 TIME COMPLEXITY ANALYSIS

**Analysis 1: Time Complexity of Matrix Entropy**

The computation of Matrix Entropy involves several complex steps, with the key bottleneck being Singular Value Decomposition (SVD), which is central to computing eigenvalues. The following steps primarily contribute to the time complexity:

| BENCHMARKS | INDICATORS | GPT MODEL SIZE | | | | | | |
|---|---|---|---|---|---|---|---|---|
| | | 111M | 256M | 590M | 1.3B | 2.7B | 6.7B | 13B |
| OPENBOOKQA | ACCURACY | 0.118 | 0.158 | 0.158 | 0.166 | 0.206 | 0.238 | 0.286 |
| | MATRIX ENTROPY | 0.3575 | 0.3416 | 0.3237 | 0.3140 | 0.2991 | 0.2848 | 0.2767 |
| | LOSS | 5.6196 | 5.3536 | 5.1881 | 4.9690 | 4.8723 | 4.7195 | 4.7050 |
| | PPL | 148.38 | 108.10 | 83.45 | 65.10 | 50.93 | 41.80 | 40.89 |
| | MATRIX NUCLEAR-NORM | 0.4447 | 0.4057 | 0.3941 | 0.3644 | **0.4606** | 0.3672 | **0.4423** |
| WINOGRANDE | ACCURACY | 0.488 | 0.511 | 0.498 | 0.521 | 0.559 | 0.602 | 0.646 |
| | MATRIX ENTROPY | 0.4073 | 0.3915 | 0.3706 | 0.3605 | 0.3419 | 0.3272 | 0.3149 |
| | LOSS | 4.7869 | 4.5854 | 4.4141 | 4.2513 | 4.1107 | 4.0109 | 4.0266 |
| | PPL | 39.81 | 30.25 | 26.57 | 21.87 | 18.55 | 16.53 | 16.94 |
| | MATRIX NUCLEAR-NORM | 0.4802 | 0.4479 | 0.4440 | 0.4133 | **0.5232** | 0.4220 | **0.4964** |
| PIQA | ACCURACY | 0.594 | 0.613 | 0.627 | 0.664 | **0.701** | 0.739 | **0.766** |
| | MATRIX ENTROPY | 0.4168 | 0.3991 | 0.3783 | 0.3676 | 0.3504 | 0.3344 | 0.3264 |
| | LOSS | 4.8425 | 4.5470 | 4.4029 | 4.1613 | 4.0075 | 3.8545 | 3.8826 |
| | PPL | 69.80 | 47.94 | 37.88 | 28.76 | 23.15 | 19.76 | 19.72 |
| | MATRIX NUCLEAR-NORM | 0.4868 | 0.4327 | 0.4164 | 0.3826 | **0.4452** | 0.3675 | **0.4149** |

Table 9: Language modeling indicators on openbookqa, winogrande and piqa.Except for the matrix nuclear norm, the data is sourced from Wei et al. (2024)

Table 10: The table illustrates the performance metrics for a range of GPT models on the Dolly-15k, Wikipedia, OpenWebText2, and HH-RLHF datasets, encompassing matrix entropy, loss, and perplexity. Except for the matrix nuclear norm, the data is sourced from Wei et al. (2024), underscoring the relationship between model scale and its performance.

| DATASET | INDICATORS | GPT MODELS SIZE | | | | | | |
|---|---|---|---|---|---|---|---|---|
| | | 111M | 256M | 590M | 1.3B | 2.7B | 6.7B | 13B |
| DOLLY-15K | MATRIX ENTROPY | 0.5976 | 0.5840 | 0.5582 | 0.5477 | 0.5240 | 0.5064 | 0.4859 |
| | LOSS | 3.6710 | 3.2907 | 3.0359 | 2.7517 | 2.5015 | 2.2911 | 2.3098 |
| | PPL | 39.93 | 27.53 | 21.42 | 16.15 | 12.50 | 10.23 | 10.30 |
| | MATRIX NUCLEAR-NORM | 0.6207 | 0.5565 | 0.5063 | 0.4553 | 0.4639 | 0.3904 | 0.4859 |
| WIKIPEDIA | MATRIX ENTROPY | 0.6177 | 0.6077 | 0.5848 | 0.5786 | 0.5523 | 0.5368 | 0.5126 |
| | LOSS | 3.2900 | 2.9343 | 2.6854 | 2.4282 | 2.2045 | 2.0216 | 2.0327 |
| | PPL | 31.38 | 22.51 | 17.89 | 13.85 | 11.08 | 9.19 | 9.32 |
| | MATRIX NUCLEAR-NORM | 0.6744 | 0.6422 | 0.6094 | 0.5639 | 0.5438 | 0.4660 | 0.4708 |
| OPENWEBTEXT2 | MATRIX ENTROPY | 0.6527 | 0.6479 | 0.6206 | 0.6142 | 0.5855 | 0.5683 | 0.5463 |
| | LOSS | 3.7509 | 3.3852 | 3.1414 | 2.8860 | 2.6465 | 2.4708 | 2.4685 |
| | PPL | 36.79 | 25.82 | 20.34 | 15.89 | 12.51 | 10.57 | 10.51 |
| | MATRIX NUCLEAR-NORM | 0.7147 | 0.7066 | 0.6823 | 0.6363 | 0.6017 | 0.5133 | 0.4991 |
| HH-RLHF | MATRIX ENTROPY | 0.5753 | 0.5635 | 0.5350 | 0.5268 | 0.4971 | 0.4813 | 0.4640 |
| | LOSS | 3.3078 | 2.9964 | 2.8171 | 2.6431 | 2.4622 | 2.3526 | 2.3323 |
| | PPL | 18.97 | 14.01 | 11.62 | 9.73 | 8.12 | 7.27 | 7.19 |
| | MATRIX NUCLEAR-NORM | 0.6309 | 0.5716 | 0.5307 | 0.4771 | 0.4959 | 0.4277 | 0.4518 |

1. **Matrix Normalization**: This step has a time complexity of $O(m \cdot n)$, where $m$ is the number of rows and $n$ is the number of columns.

2. **Computing the Inner Product Matrix**: Calculating $Z^T Z$ has a time complexity of $O(n^2 \cdot m)$ due to the multiplication of two matrices sized $m \times n$.

3. **Singular Value Decomposition (SVD)**: The time complexity of SVD is $O(n^3)$, which is the primary computational bottleneck, especially for large $n$.

Therefore, the total time complexity of Matrix Entropy can be approximated as:

$$O(m \cdot n + n^2 \cdot m + n^3) = O(n^3)$$

This complexity indicates that Matrix Entropy becomes increasingly impractical for large-scale models as $n$ grows.

**Analysis 2: Time Complexity of Matrix Nuclear-Norm**

Table 11: Language modeling indicators for Pythia models across Dolly-15k, Wikipedia, OpenWeb-Text2, and HH-RLHF datasets (lower values indicate better performance). Except for the matrix nuclear norm, data is derived from Wei et al. (2024), showcasing the correlation between model scale and performance.

| DATASETS | INDICATORS | PYTHIA MODELS SIZE | | | | | | | | | |
|---|---|---|---|---|---|---|---|---|---|---|---|
| | | 14M | 31M | 70M | 160M | 410M | 1B | 1.4B | 2.8B | 6.9B | 12B |
| DOLLY-15K | MATRIX ENTROPY | 0.7732 | 0.7155 | 0.6707 | 0.6243 | 0.5760 | 0.5328 | 0.5309 | 0.5263 | 0.5003 | 0.4876 |
| | LOSS | 4.4546 | 4.0358 | 3.5990 | 3.1323 | 2.6752 | 2.4843 | 2.3816 | 2.2484 | 2.1368 | 2.0616 |
| | MATRIX NUCLEAR-NORM | 0.7508 | 0.7735 | 0.6984 | 0.6104 | 0.5760 | 0.4710 | 0.4922 | 0.4585 | 0.4202 | 0.4181 |
| WIKIPEDIA | MATRIX ENTROPY | 0.7938 | 0.7442 | 0.7003 | 0.6580 | 0.6039 | 0.5584 | 0.5587 | 0.5553 | 0.5314 | 0.5140 |
| | LOSS | 4.1112 | 3.6921 | 3.2694 | 2.8207 | 2.4017 | 2.2213 | 2.0140 | 1.9120 | 1.8489 | |
| | MATRIX NUCLEAR-NORM | 0.6053 | 0.6700 | 0.6996 | 0.6718 | 0.6464 | 0.5591 | 0.5787 | 0.5410 | 0.4850 | 0.4768 |
| OPENWEBTEXT2 | MATRIX ENTROPY | 0.8144 | 0.7749 | 0.7370 | 0.6980 | 0.6415 | 0.5944 | 0.5916 | 0.5887 | 0.5591 | 0.5417 |
| | LOSS | 4.3965 | 4.0033 | 3.6284 | 3.2031 | 2.7838 | 2.6198 | 2.5228 | 2.4005 | 2.3133 | 2.2502 |
| | MATRIX NUCLEAR-NORM | 0.5041 | 0.6186 | 0.7142 | 0.7258 | 0.7105 | 0.6215 | 0.6378 | 0.5967 | 0.5275 | 0.5110 |
| HH-RLHF | MATRIX ENTROPY | 0.7673 | 0.7114 | 0.6607 | 0.6126 | 0.5552 | 0.5054 | 0.5032 | 0.4977 | 0.4699 | 0.4528 |
| | LOSS | 3.7466 | 3.4018 | 3.1146 | 2.7366 | 2.4340 | 2.3311 | 2.2687 | 2.1992 | 2.1199 | 2.0905 |
| | MATRIX NUCLEAR-NORM | 0.7353 | 0.7674 | 0.6964 | 0.6182 | 0.5886 | 0.4825 | 0.5141 | 0.4839 | 0.4562 | 0.4481 |

| Prompt ID | Prompt Content |
|---|---|
| Prompt 1 | You are an AI assistant. You will be given a task. You must generate a detailed and long answer. |
| Prompt 2 | You are a helpful assistant, who always provide explanation. Think like you are answering to a five year old. |
| Prompt 3 | You are an AI assistant. User will give you a task. Your goal is to complete the task as faithfully as you can. While performing the task think step-by-step and justify your steps. |

Table 12: The prompts selected from OpenOrca(Lian et al., 2023b) dataset.

Matrix Nuclear-Norm avoids the SVD step by approximating the nuclear norm using the L2 norm, resulting in a more efficient computation. The analysis is as follows:

1. **Matrix Normalization**: Similar to Matrix Entropy, this step has a time complexity of $O(m \cdot n)$.

2. **Calculating the L2 Norm**: For each column vector, the L2 norm is computed with a complexity of $O(m \cdot n)$, where we take the square root of the sum of squares for each column vector.

3. **Sorting and Extracting the Top D Features**: Sorting the L2 norms has a complexity of $O(n \log n)$.

Therefore, the overall time complexity of Matrix Nuclear-Norm is:

$$O(m \cdot n + n \log n) \approx O(n^2) \quad \text{when} \quad m \approx n$$

This indicates that Matrix Nuclear-Norm is computationally more efficient, especially as $n$ increases.

A.4.2 EXPERIMENTAL VALIDATION AND COMPARATIVE ANALYSIS

To empirically validate the theoretical time complexities, we conducted experiments using matrices of various sizes. Figure 7 shows that as $n$ increases, Matrix Nuclear-Norm consistently outperforms Matrix Entropy in terms of runtime, confirming the theoretical advantage.

**Discussion of Assumptions and Applicability** Our complexity analysis assumes $m \approx n$, which holds in many real-world applications, such as evaluating square matrices in large-scale language models. However, in cases where $m \neq n$, the time complexity might differ slightly. Nonetheless, Matrix Nuclear-Norm is expected to maintain its efficiency advantage due to its avoidance of the costly SVD operation.

**Impact of Constant Factors** Although both $O(n^2)$ and $O(n^3)$ indicate asymptotic behavior, Matrix Nuclear-Norm's significantly smaller constant factors make it computationally favorable even for moderately sized matrices, as evidenced in our experimental results.

A.4.3 CONCLUSION OF THE COMPLEXITY ANALYSIS

Through this detailed analysis and experimental validation, we conclude the following:

- Matrix Entropy, with its reliance on SVD, has a time complexity of $O(n^3)$, making it computationally expensive for large-scale applications.

| LENGTH | GPT MODEL SIZE | | | | | | |
|---|---|---|---|---|---|---|---|
| | 111M | 256M | 590M | 1.3B | 2.7B | 6.7B | 13B |
| 64 | 0.4574 | 0.4125 | 0.3787 | 0.3486 | 0.4053 | 0.3315 | 0.4148 |
| 128 | 0.5293 | 0.4680 | 0.4270 | 0.3835 | 0.4143 | 0.3477 | 0.4032 |
| 512 | 0.7883 | 0.6978 | 0.6251 | 0.5554 | 0.5265 | 0.4468 | 0.4422 |
| 1024 | 0.9132 | 0.8787 | 0.7802 | 0.6953 | 0.6351 | 0.5383 | 0.5028 |

Table 13: Analysis of Length Dynamics

- Matrix Nuclear-Norm, by using the L2 norm approximation, achieves a time complexity of $O(m \cdot n + n \log n) \approx O(n^2)$, significantly reducing computational costs.
- Experimental results confirm that Matrix Nuclear-Norm offers superior time efficiency for evaluating large-scale models, particularly those with millions or billions of parameters.

### A.5 PROOF OF THEOREM 1

This section presents the proof of the strictly inverse monotonic relationship between the entropy H(A) and the Frobenius norm $\|A\|_F$ for a matrix $A$.

Let $A \in \mathbb{R}^{B \times C}$ be a non-negative matrix where each row represents a probability distribution:

$$\sum_{j=1}^{C} A_{i,j} = 1, \quad \forall i = 1, 2, \ldots, B$$

with $A_{i,j} \geq 0$. Here, $A_{i,j}$ denotes the predicted probability that sample $i$ belongs to category $j$.

The Shannon entropy $H(A)$ of the matrix $A$ is defined as:

$$H(A) = -\frac{1}{B} \sum_{i=1}^{B} \sum_{j=1}^{C} A_{i,j} \log(A_{i,j})$$

where $0 \log(0)$ is defined as $0$ by convention.

The Frobenius norm $\|A\|_F$ is defined as:

$$\|A\|_F = \sqrt{\sum_{i=1}^{B} \sum_{j=1}^{C} A_{i,j}^2}.$$

**Step 1: Entropy and Frobenius Norm for a Single Row**

Consider a single row $\mathbf{a} = [a_1, a_2, \ldots, a_C]$, where $a_j = A_{i,j}$, $a_j \geq 0$, and $\sum_{j=1}^{C} a_j = 1$. The row entropy is:

$$H_i = -\sum_{j=1}^{C} a_j \log(a_j),$$

and the row Frobenius norm is:

$$\|\mathbf{a}\|_2 = \sqrt{\sum_{j=1}^{C} a_j^2}.$$

To determine the extrema of $H_i$, we use the method of Lagrange multipliers. Define the Lagrangian:

$$L(a_1, a_2, \ldots, a_C, \lambda) = -\sum_{j=1}^{C} a_j \log(a_j) + \lambda \left( \sum_{j=1}^{C} a_j - 1 \right).$$

Taking the partial derivatives and setting them to zero:

$$\frac{\partial L}{\partial a_j} = -\log(a_j) - 1 + \lambda = 0 \implies a_j = e^{\lambda - 1}.$$

Using the normalization condition $\sum_{j=1}^{C} a_j = 1$, we find $a_j = \frac{1}{C}$ for all $j$. Substituting $a_j = \frac{1}{C}$ into $H_i$ and $\|\mathbf{a}\|_2$:

$$H_i = \log(C), \quad \|\mathbf{a}\|_2 = \sqrt{\frac{1}{C}}.$$

For the minimum entropy, let $a_k = 1$ and $a_j = 0$ for $j \neq k$:

$$H_i = 0, \quad \|\mathbf{a}\|_2 = 1.$$

Thus, $H_i$ and $\|\mathbf{a}\|_2$ exhibit an inverse monotonic relationship.

**Step 2: Generalizing to the Entire Matrix**
The matrix-level entropy $H(A)$ and Frobenius norm $\|A\|_F$ are given by:

$$H(A) = \frac{1}{B} \sum_{i=1}^{B} H_i, \quad \|A\|_F = \sqrt{\sum_{i=1}^{B} \|\mathbf{a}_i\|_2^2}$$

Since $H(A)$ is the average of row entropies and $\|A\|_F$ is derived from the sum of row Frobenius norms, the inverse monotonicity established for a single row generalizes to the entire matrix.

**Step 3: Bounds for $\|A\|_F$**
To determine the bounds for $\|A\|_F$:
Maximum Frobenius Norm: When each row is a one-hot vector (minimum entropy):

$$\|A\|_F = \sqrt{\sum_{i=1}^{B} 1} = \sqrt{B}$$

Minimum Frobenius Norm: When each row is uniformly distributed (maximum entropy):

$$\|A\|_F = \sqrt{\sum_{i=1}^{B} \sum_{j=1}^{C} \left(\frac{1}{C}\right)^2} = \sqrt{\frac{B}{C}}$$

**Step 4: Implications for Model Evaluation**
The inverse monotonic relationship between $H(A)$ and $\|A\|_F$ implies that models with higher $\|A\|_F$ exhibit greater discriminability and certainty in their predictions. This makes $\|A\|_F$ a useful proxy for evaluating the compression and confidence capabilities of large language models (LLMs).

**Conclusion**
The proof establishes that $H(A)$ and $\|A\|_F$ are strictly inversely monotonic. This relationship provides theoretical justification for using $\|A\|_F$ as an evaluation metric in LLMs, where balancing diversity and confidence is essential.

A.6    PROOF OF THEOREM 3

Given that $\|A\|_F \approx \sqrt{B}$, we approximate the $j$-th largest singular value $\sigma_j$ as top $\left(\sum_{i=1}^{B} A_{i,j}^2, j\right)$. This result is derived by analyzing the contributions of $A$'s columns.

**1. Decomposition of $A$ and the Gram Matrix:** Using the Singular Value Decomposition (SVD), $A = U\Sigma V^T$, where $\Sigma$ is a diagonal matrix containing the singular values $\sigma_1, \sigma_2, \ldots, \sigma_D$, with $D = \min(B, C)$. The Gram matrix $A^T A$ has diagonal elements given by:

$$(A^T A)_{j,j} = \sum_{i=1}^{B} A_{i,j}^2,$$

which represents the squared norm of the $j$-th column of $A$.

**2. Connecting Column Norms to Singular Values:** Singular values measure the contributions of orthogonal projections of $A$. When $\|A\|_F \approx \sqrt{B}$, most contributions to the nuclear norm $\|A\|_*$ come from the largest column norms $\sqrt{\sum_{i=1}^{B} A_{i,j}^2}$.

**3. Approximation of Singular Values:** For matrices with well-distributed entries in their columns, the top singular values $\sigma_j$ approximately correspond to the largest column norms. Therefore, for $j \in \{1, \ldots, D\}$:

$$\sigma_j \approx \text{top}\left(\sqrt{\sum_{i=1}^{B} A_{i,j}^2}, j\right).$$

**4. Efficient Approximation of Nuclear Norm:** Using this approximation, the batch nuclear norm can be efficiently computed as:

$$\|\hat{A}\|_* = \sum_{j=1}^{D} \text{top}\left(\sqrt{\sum_{i=1}^{B} A_{i,j}^2}, j\right).$$

Here, $\text{top}(\cdot, j)$ denotes the $j$-th largest value in the set. This approximation assumes that $A$'s entries are approximately well-distributed across columns, a condition commonly satisfied when $\|A\|_F \approx \sqrt{B}$.

