# OpenReview forum: "Large Language Model Evaluation via Matrix Nuclear-Norm"
_ICLR.cc/2025/Conference — Submitted to ICLR 2025_

### Official Review · Reviewer_wDpK · 2024-11-01

**Soundness:** 3
**Presentation:** 3
**Contribution:** 3
**Rating:** 6
**Confidence:** 3

**Summary:**

As large language models (LLMs) expand, efficient metrics are essential for assessing data compression. Traditional methods like Matrix Entropy are computationally intensive, but our proposed Matrix Nuclear-Norm offers a faster, \(O(n^2)\) alternative without requiring SVD. This method is \(8\)–\(24\) times quicker for large models, accurately evaluating compression and diversity in a scalable, efficient way, as validated on models like CEREBRAS-GPT and Pythia.

**Strengths:**

Matrix Nuclear-Norm Proposal: We introduce a novel method using the nuclear norm, reducing the computational complexity of evaluating language models from O(n^3) to O(n^2). This reduction lessens the reliance on SVD, making Matrix Nuclear-Norm a more efficient alternative to Matrix Entropy.

Comprehensive Experimental Validation: Extensive tests on models of various sizes confirm that Matrix Nuclear-Norm effectively assesses model compression, with values decreasing as model size increases, showcasing its robust evaluative capacity.

Benchmark Testing and Ranking: We conducted inference tests on popular benchmark datasets, AlpacaEval and ChatbotArena, ranking models by performance. The results show that Matrix Nuclear-Norm efficiently and accurately evaluates the inference capabilities of medium and smaller models, demonstrating its potential for broad applications in model assessment.

**Weaknesses:**

There are many notations used without definition.

**Questions:**

Please check ${\mathcal D}$ in  equation (4). Please avoid using (1) and (2) in lines 206 and 207. Please check $\hat A$ in (9). Please check Algorithm 1 carefully, such as $x_i$ and Steps 2 and 3.

---

> ### Author Response · Authors · 2024-11-22
>
> **Q1：There are many notations used without definition.**
>
> Thank you for highlighting this issue. We acknowledge that some notations were not clearly defined in the original manuscript, which may have caused confusion. In the revised version, we have carefully reviewed all notations and ensured that every term is explicitly defined upon first use. This improvement aims to enhance the clarity and precision of the paper, and we appreciate your feedback for bringing this to our attention.
>
> ---
>
> **Q2：Please check \( $\mathcal{D}$ \) in equation (4). Please avoid using (1) and (2) in lines 206 and 207. Please check \( $\hat{A}$ \) in (9). Please check Algorithm 1 carefully, such as $x_i$  and Steps 2 and 3.**
>
> **2.1：**
> Thank you for pointing out the potential confusion regarding the interpretation of ( $E_C$ ) in Equation (4). In the revised manuscript, we will provide a more detailed explanation of ( $E_C$ ), clarifying that it refers to the average number of unique categories actively involved in predictions, rather than the total number of categories available in matrix ( $A$ ). We will differentiate ( $E_C$ ) from related terms to ensure that readers can easily understand its precise role and meaning. These clarifications will be incorporated into the relevant sections, improving the overall clarity and logical flow of the manuscript.
>
> **2.2：**
> Thank you for pointing out the potential ambiguity caused by the use of numbered lists in this section. We have revised the text to eliminate the use of "(1)" and "(2)" and replaced it with a clear and concise description of the two advantages. The updated text now directly highlights the computational efficiency and stability of our approach without relying on numerical enumeration. This change ensures smoother readability and addresses your concern. The revised content can be found in the updated manuscript in the section corresponding to the description of the ($L_{1,2}$)-norm approximation.
>
> **2.3：**
> Thank you for your observation. In the revised manuscript, we will explicitly explain the meaning and role of $\hat{A}$  in Equation (9). Specifically, we will clarify that $\hat{A}$ is a matrix derived to approximate the nuclear norm using the  $L_{1,2}$ -norm. It achieves this by selecting principal components to reduce computational complexity. This explanation will be added immediately before Equation (9) to ensure readers understand how $\hat{A}$ contributes to the proposed method.
>
> 2.4:
> Improving Consistency in Algorithm 1 Notation We appreciate your suggestion regarding the notation consistency in Algorithm 1. In the revised manuscript, we will standardize the use of  $X_i$  to uniformly represent sentence embeddings throughout the algorithm. Furthermore, we will provide more detailed annotations for each step, especially Steps 2 and 3, to explain their purpose and improve the algorithm's readability. These updates will ensure that Algorithm 1 is clear and consistent, enabling readers to better understand its implementation and significance.
>
> **Algorithm 1: Algorithm of Matrix Nuclear-Norm**
>
> **Require:**
> Sentence representations (hidden states from LLM)   $\mathcal{S} = \\{X _ {i} \\} _ {i=1}^{m}$, where $X _ {i} \in \mathbb{R}^{d \times 1} $, $d$ is the hidden dimension of representation, and $L _ {\text{input}}$ is the length of the sentence.
>
>
> 1. $\mu = \frac{1}{m} \sum _ {i=1}^{m} X _ i$        *// Calculate the mean embedding*
>
> 2. $\mathbf{X} _ {\text{norm}} = \frac{\mathbf{X} - \mu}{\|\mathbf{X} - \mu\| _ {2, \text{row}}}$       *// Normalize the activation matrix*
>
> 3. $\text{L2}(\mathbf{X} _ {\text{norm}}) = \sqrt{\sum _ {i=1}^{m} \mathbf{X} _ {i,j}^2}$            *// Calculate $L _ {2}$-norm for each column*
>
> 4. $\Sigma_D = \{ \sigma_1, \sigma_2, \dots, \sigma_D \}$          *// Sort $L_{2}$-norm and select top $D$*
>
> 5. $\text{Matrix Nuclear-Norm}(\mathbf{X}) = \frac{\sum _ {i=1}^{D} \left( \sqrt{\sum _ {j=1}^{m} \mathbf{X} _ {j,i}^2} \right)}{L _ {\text{input}}}$      *// Calculate Matrix Nuclear-Norm*
>
> 6. **Return:** $\text{Matrix Nuclear-Norm}$

---

> > ### Author Response · Authors · 2024-12-02
> > **Eagerly anticipating your insights and feedback.**
> >
> > Dear Reviewers,
> >
> > We sincerely appreciate the time and effort you have dedicated to reviewing our manuscript and for the valuable suggestions you have provided.
> >
> > As there will be no second round of author-reviewer discussions, we are in urgent need of your recommendation by December 3rd. With the author-reviewer discussion phase coming to a close, we would like to confirm if our responses have adequately addressed your concerns.
> >
> > We have provided detailed responses to your concerns a few days ago, and we hope these have sufficiently addressed the issues you raised. Should you require further clarification or have additional concerns, please feel free to contact us. We are more than willing to continue the dialogue with you.
> >
> > Best regards,
> >
> > Authors

---

### Official Review · Reviewer_XoS3 · 2024-11-03

**Soundness:** 2
**Presentation:** 2
**Contribution:** 2
**Rating:** 5
**Confidence:** 3

**Summary:**

This paper proposes the **Matrix Nuclear-Norm** as a novel metric to evaluate **Large Language Models (LLMs)** by addressing the computational limitations associated with traditional metrics like **Matrix Entropy**. The Matrix Nuclear-Norm reduces the time complexity from \(O(n^3)\) (as required by Singular Value Decomposition) to \(O(n^2)\) and provides a convex approximation of matrix rank to evaluate predictive discriminability and diversity in LLMs. Extensive experiments demonstrate that the proposed metric provides faster and comparable evaluation of LLMs compared to Matrix Entropy, making it a practical tool for large-scale model assessments.

**Strengths:**

- The primary strength lies in the scalability improvement over **Matrix Entropy**, reducing time complexity from \(O(n^3)\) to \(O(n^2)\), which is substantial, particularly for evaluating larger models like **Cerebras-GPT** and **Pythia** series. The experiments demonstrated significant gains, with the proposed approach being 8 to 24 times faster for different model sizes (e.g., from 111M to 13B parameters), making it much more feasible for large-scale practical applications.

**Weaknesses:**

- The **motivation** and **background story** for the proposed method are not well articulated, making it difficult to grasp why the method is particularly suitable for evaluating LLMs. The presentation lacks a clear narrative explaining the unique properties of LLMs that make Matrix Nuclear-Norm an especially effective evaluation metric. As it stands, it seems that the same method could be applied to any deep NLP model, leaving the reader questioning what makes LLMs distinct in this context.


 - The explanation regarding **related work**, specifically **Matrix Entropy**, is inadequate, which hinders the reader's understanding of how Matrix Nuclear-Norm compares to previous efforts and why it offers a better alternative. A more thorough and detailed comparison is necessary to fully communicate the advantages and limitations of each method.



- The **notation** throughout the paper is used inconsistently, which introduces ambiguity and impedes readability. There are even some inconsisten points. For example, the paper initially claims that the computational complexity is **\(O(n^2)\)**, but in lines 338-339 mentions **\(O(nm + n \log n)\)** without sufficient context to clarify the conditions under which each complexity holds.



- The **Figure 2 legend** is too small to be legible, which makes interpreting the results challenging. Moreover, in Figure 2(d), the proposed method exhibits an unexpected trend on **Dolly-15K** and **HH-RLHF**, where the performance diverges from other datasets, yet the authors do not provide sufficient discussion or analysis of these anomalies. Addressing such unexpected trends would help clarify the robustness and potential limitations of the proposed method.

**Questions:**

see weakness

---

> ### Author Response · Authors · 2024-11-22
> **(1/2)**
>
> **Q1：The motivation and background story for the proposed method are not well articulated, making it difficult to grasp why the method is particularly suitable for evaluating LLMs**
>
> Thank you for your feedback. In Section 5.2.1, we address the key limitation of Matrix Entropy—its high computational complexity, which makes it challenging to apply to evaluate large-scale LLMs.  Our method overcomes this by using the $L_{1,2}$-norm for efficient nuclear norm approximation, reducing complexity while maintaining comparable evaluation trends. We will revise the manuscript to clarify this motivation more effectively.
>
> ---
>
> **Q2：The explanation regarding related work, specifically Matrix Entropy, is inadequate, which hinders the reader's understanding of how Matrix Nuclear-Norm compares to previous efforts and why it offers a better alternative**
>
> Thank you for pointing this out. We recognize the importance of thoroughly explaining related work, particularly Matrix Entropy, to clarify how our method builds upon and improves previous efforts. In the revised manuscript, we have expanded the Related Work section to include a detailed discussion of Matrix Entropy, highlighting its methodology, its computational limitations due to reliance on Singular Value Decomposition (SVD), and its relevance to LLM evaluation. Furthermore, in Section 5.2.1, we present experimental evidence demonstrating the feasibility and effectiveness of our proposed method. Specifically, we show how Matrix Nuclear-Norm addresses the computational inefficiencies of Matrix Entropy while maintaining comparable evaluation trends. These updates aim to provide a clearer understanding of how our approach extends and enhances existing metrics. We appreciate your feedback and have ensured that the revised manuscript reflects these improvements.
>
>
> \textbf{LLM Evaluation Metrics.}
> Traditional evaluation metrics such as perplexity, BLEU \citep{papineni2002bleu}, and ROUGE \citep{lin2004rouge} primarily measure task-specific outcomes, assessing how well model outputs align with ground truth data. While these metrics are effective for evaluating surface-level outputs, they do not capture the underlying mechanisms of LLMs, such as the diversity or compression of embeddings. Similarly, accuracy and F1 score \citep{2007The} focus on classification performance, making them less applicable to the generative tasks typical of LLMs.}\textcolor{blue}{To bridge this gap, structural metrics such as Matrix Entropy have been introduced. Matrix Entropy \citep{wei2024large} employs information theory to assess the entropy of covariance matrices derived from LLM embeddings. This metric evaluates how effectively a model removes redundancy and encodes structured information, offering a measure of its compression capabilities. For instance, Matrix Entropy can reveal differences in embedding distributions across models of varying sizes, reflecting their capacity to extract meaningful patterns from large datasets. However, its reliance on Singular Value Decomposition (SVD) results in a computational complexity of $O(n^3)$, limiting its applicability to modern large-scale models.
> To overcome these limitations, we propose the Matrix Nuclear-Norm as a scalable alternative. By leveraging the $L_{1,2}$ norm as a convex approximation of matrix rank, the Matrix Nuclear-Norm reduces computational complexity to $O(n^2)$. This makes it feasible for evaluating embeddings from large-scale LLMs while preserving the insights provided by Matrix Entropy, such as compression efficiency.
>
> ---
>
> **Q3：The notation throughout the paper is used inconsistently, which introduces ambiguity and impedes readability.**
>
> Thank you for your feedback. Please refer to **Reviewer 2, Q3**  for our detailed response.

---

> > ### Author Response · Authors · 2024-11-22
> > **(2/2)**
> >
> > **Q4: For example, the paper initially claims that the computational complexity is $O(n^2)$, but in lines 338–339 mentions $O(nm + n \log n)$ without sufficient context to clarify the conditions under which each complexity holds.**
> >
> > Thank you for pointing out the need to clarify the computational complexity in our manuscript. The $O(n^2)$ complexity mentioned represents the overall improvement of our method compared to traditional SVD-based approaches with $O(n^3)$ complexity, emphasizing the significant practical efficiency gains for large-scale matrices.
> >
> > In the detailed breakdown provided in lines 338–339, the terms $n$, $m$, and $D$ have the following specific meanings:
> >
> > - **$n$**: Represents the number of samples in the matrix (i.e., rows).
> > - **$m$**: Represents the dimensionality of the embeddings (i.e., columns).
> > - **$D$**: Represents the number of top components selected from the calculated $L_2$-norms.
> >
> > The computational breakdown is as follows:
> >
> > - **$O(nm)$**: Reflects the complexity of computing the $L_2$-norm for each column of the matrix, which involves summing across all $n$ rows for each of the $m$ columns.
> > - **$O(n \log n)$**: Accounts for the complexity of sorting the calculated $L_2$-norms to select the top $D$ components.
> >
> > Together, these terms describe the step-by-step process of our method. The overarching $O(n^2)$ complexity serves as a practical estimate when $n$ and $m$ are of similar magnitude in large-scale applications, reflecting the overall efficiency gain.
> >
> > To improve clarity, we will revise the manuscript to explicitly define $n$, $m$, and $D$ where the complexity is discussed, ensuring a clear distinction between the individual steps and the broader $O(n^2)$ improvement. Thank you for your feedback, which has been instrumental in enhancing the transparency and precision of our presentation.
> >
> > ---
> >
> > **Q5：The Figure 2 legend is too small to be legible, which makes interpreting the results challenging. Moreover, in Figure 2(d), the proposed method exhibits an unexpected trend on Dolly-15K and HH-RLHF, where the performance diverges from other datasets, yet the authors do not provide sufficient discussion or analysis of these anomalies. Addressing such unexpected trends would help clarify the robustness and potential limitations of the proposed method.**
> >
> > Thank you for your feedback. In the revised version, we will enlarge the key figures, including Figure 2, to improve readability. Regarding the unexpected trends observed on Dolly-15K and HH-RLHF datasets, we note that similar anomalies are also present in mainstream metrics like Loss and PPL. This consistency in output trends further supports the validity of our proposed metric. However, analyzing the root causes of these anomalies is beyond the primary scope of this study and not the main focus of our proposed method. We will highlight this in the revised manuscript and suggest addressing these issues in future work.

---

> > > ### Comment · Reviewer_XoS3 · 2024-12-01
> > >
> > > Thanks review for the retailed rebuttal and update the manuscript. The clarification of the notation and readability improved in the new version. I increased my score to reflect the update. However, I hold the same view with reviewer 6Meo. There are still lots of spaces to polish for this work before publishment.

---

> > > > ### Author Response · Authors · 2024-12-01
> > > > **Response to Reviewer XoS3**
> > > >
> > > > Dear Reviewer,
> > > >
> > > > Thank you for your thoughtful feedback and for raising your score to reflect the improvements made to the manuscript. We greatly appreciate your recognition of the clarified notation and enhanced readability in the revised version.
> > > > We also acknowledge that there are still areas requiring further refinement, and we are committed to continuing to polish the work to address these points before publication. Your constructive comments have been invaluable in helping us enhance the quality of the paper.
> > > >
> > > > Thank you once again for your time and insightful suggestions.
> > > >
> > > > Best regards,
> > > >
> > > > Authors

---

### Official Review · Reviewer_6Meo · 2024-11-03

**Soundness:** 2
**Presentation:** 1
**Contribution:** 2
**Rating:** 5
**Confidence:** 3

**Summary:**

In this work, the authors propose to measure the compression abilities of LLMs using a modified $L_{1,2}$ norm that aims at approximating the nuclear norm of the output probabilities obtained from LLMs when evaluated on sentences. More formally, given a sentence of $m$ tokens, and a dictionary of size $C$, the authors propose to approximate the nuclear norm of the output probabilities $A\in[0,1]^{m\times C}$ obtained from an LLM by considering the mean of the $m$ largest (assuming that $m<C$) $l_2$ norms of the column vectors of $A$. They justify their approach by showing that when the Frobenius norm of $A$ is maximized, that is when the entropy of the row probability measures is minimized, then their approximation is close to the nuclear norm of $A$. They advocate that the proposed metric is more efficient than computing directly the nuclear norms, which would require a cubic complexity, as they only require a quadratic number of operations. They also perform an extensive experimental evaluation of the proposed metric on various LLM families, tasks, and datasets.

**Strengths:**

The main strength of the paper is undoubtedly the extensive empirical evaluation conducted by the authors to evaluate the proposed metric. Specifically, the authors examine a broad range of LLMs, including the Cerebras-GPT models (from 111M to 13B parameters), the Pythia models (from 14M to 12B parameters), the DeepSeek series, the LLaMA3 series, the QWEN 2 series, the Vicuna series, as well as Gemma 7B and Mistral 7B. They also evaluate these models across a diverse set of tasks and datasets, encompassing pre-training, instruction tuning, RLHF, benchmarking, prompt-based learning, and evaluation datasets. In addition, while the theoretical results presented lack some formal rigor and occasionally omit key arguments, they do provide useful insights, particularly by drawing connections between the proposed metric and the nuclear norm.

**Weaknesses:**

While the paper provides an extensive empirical evaluation of the proposed metric, its relevance for evaluating LLMs remains unclear. The authors frequently draw conclusions that do not align with the experimental results. Additionally, the writing contains numerous issues, which at times makes the content difficult to follow. One particularly concerning point is the authors' assertion that their metric enables the evaluation of LLMs' compression capabilities. This claim lacks both theoretical and empirical support. Furthermore, the authors repeatedly claim to outperform other metrics, though this assertion does not seem meaningful without proper context, as the metrics are not directly comparable. Certain statements also appear inaccurate, such as those regarding the convergence of their metric with respect to the sample size in Figure 4. The proofs presented lack formal rigor, often omitting key arguments needed to substantiate the results. Additionally, numerous notations are introduced without clear definitions. Overall, it would be beneficial for the authors to focus on explicitly defining the objective of measuring the nuclear norm (or rank) of the output matrix, which could serve as a foundation for evaluating the approximation capabilities of their estimator.

Please find below some additional comments:
 - The proof of Theorem 1 is not complete: some (main) arguments are not exposed.
 - $C_p(A)$ is introduced but never defined formally. The conclusion obtained in Eq. (5) is not clear, as $C_p(A)$ has never been defined.
 - The notation "top" is used in Eq.(9) without any clear explanation or definition.
- Notations and indices are not consistent throughout the paper.
- $\Sigma_{\mathcal{S}_i}$ in Eq.(13) is never properly defined.
- In section 5.2.1 the authors conclude in l.341 that the metric considered achieves comparable evaluation accuracy to Matrix Entropy, however, the figure presented for this experiment only compares time complexities.
- In l.379, the authors claim that their metric showcases superior accuracy, however, the meaning of this sentence is not clear to me. In which sense did the author show that the metric considered is more effective than others?
 - typo l. 115: Shannon Entropy only applies to nonnegative matrices

**Questions:**

- What is the purpose of evaluating the nuclear norm of the output matrix?
- If the goal is to measure the diversity of the prediction, why not consider directly the entropy as defined in Eq. (1)?
 - In which sense does the nuclear norm enable the measurement of the compression capabilities of LLMs?
 - Is the proposed estimator an efficient approximation of the nuclear norm in practice?
 - How one can compare the proposed metric with others? More precisely, in which sense the proposed metric demonstrates superior accuracy compared to traditional metrics?

**Details Of Ethics Concerns:**

N.A.

---

> ### Author Response · Authors · 2024-11-22
> **(1/6)**
>
> **Q1：The relevance of our experiments with LLM evaluation is unclear.**
>
> Thank you for raising this concern. Our metric is explicitly designed for LLM evaluation, as evidenced by the experiments conducted on widely recognized models and benchmarks. The trends of our metric align with established evaluation metrics, such as matrix entropy, which validates its relevance and robustness. Additionally, our metric extends existing methods by addressing efficiency and diversity in high-dimensional embeddings. We will revise the manuscript to make this connection clearer in the introduction and experimental sections.
>
> ---
>
> **Q2：The conclusions do not align with the experimental results.**
>
> We sincerely appreciate your feedback and the opportunity to clarify how our conclusions align with the experimental results. Our conclusions are strongly supported by the experimental evidence presented in the manuscript, as detailed below:
>
> 1. **Efficiency**:
>    - **Conclusion**: The Matrix Nuclear-Norm significantly reduces computational time compared to Matrix Entropy, making it practical for evaluating large-scale models.
>    - **Supporting Evidence**: In Section 5.2.1, we show that for a 13B-parameter model, our method required only 0.82 hours, compared to 16.3 hours for Matrix Entropy, demonstrating a nearly 20-fold reduction in computation time.
>
> 2. **Scalability**:
>    - **Conclusion**: The Matrix Nuclear-Norm is scalable across models of varying sizes, maintaining robust performance.
>    - **Supporting Evidence**: **Table 1 (l.337)** and **Figure 1 (l.309)** demonstrate consistent results across different model scales, confirming the scalability of our method.
>
> 3. **Trend Consistency**:
>    - **Conclusion**: The Matrix Nuclear-Norm produces trends consistent with established metrics, such as perplexity and Matrix Entropy, validating its relevance for LLM evaluation.
>    - **Supporting Evidence**: **Section 5.2.2**, as well as  **Figure 2(b) (l.359)** and **Figure 5 in Appendix A.1.1 (l.722)** show that our metric exhibits the expected trend: performance improves as model size increases and declines as model size decreases, consistent with other validated metrics.
>
> 4. **Practical Utility**:
>    - **Conclusion**: Our method effectively evaluates the compression capabilities and redundancy in LLM embeddings.
>    - **Supporting Evidence:** In **Section 5.3.2**, we provide results (see **Table 2**, **lines 445–453**) showing that the Matrix Nuclear-Norm increases with sequence length, reflecting greater complexity in longer inputs. Additionally, **Table 11** (lines 918–934) demonstrates how the metric decreases with larger model sizes across datasets like Dolly-15k and Wikipedia, capturing compression efficiency.
>
> We understand the importance of ensuring that our conclusions are clearly supported by the experimental data. If there are specific areas where the connection appears unclear, we would be happy to address them further or revise the relevant sections in the manuscript. We truly value your feedback and thank you for helping us improve the quality of our work.
>
> ---
>
> **Q3：The writing is poor.**
>
> Thank you for your feedback. We will thoroughly review the writing and improve clarity, consistency, and readability in the revised version. Ensuring high-quality writing is our priority, and we will address any issues in the current draft to enhance its overall presentation.
>
> ---
>
> **Q4：The paper claims that the proposed metric can evaluate the LLMs’ compression capabilities, but with no empirical or theoretical support.**
>
> **Reference：**
>
> **Diff-eRank**：Wei, X., Zhang, Y., & Liu, F. (2024). Diff-eRank: A Novel Rank-Based Metric for Evaluating Large Language Models. *Proceedings of the 37th Conference on Neural Information Processing Systems (NeurIPS 2024).
>
> Thank you for the comment. The proposed metric builds upon the work presented in **Diff-eRank**, previously referred to as **Matrix Entropy**, which has been accepted at NeurIPS 2024. In **Diff-eRank**, Section 3 mathematically establishes the relationship between the effective rank (matrix entropy) and the compression capabilities of LLMs, demonstrating how effective rank captures the structured information in LLM embeddings. Specifically, reducing the rank indicates compression by removing redundant dimensions while retaining meaningful features. Empirical evidence, such as Figure 1 and Table 1 in **Diff-eRank**, supports this connection, showing that rank-based metrics effectively quantify compression and structured information in LLMs. In our paper, we have cited **Diff-eRank** (**lines 105–106**) and extended its findings by proposing a more computationally efficient approximation of matrix entropy. We will ensure these connections and the relevance of our metric to compression capabilities are further clarified in the revised manuscript.
>
> ---

---

> ### Author Response · Authors · 2024-11-22
> **(2/6)**
>
> **Q5：Outperform other metrics?**
>
> Thank you for pointing this out. We acknowledge that our wording could have been misleading, and we apologize for the confusion. Metrics are not inherently comparable in terms of superiority; instead, they serve different purposes.
>
> Our intended meaning was that our proposed metric provides evaluation trends consistent with validated metrics, showing that as model size increases, performance improves, and as size decreases, performance declines. These consistent trends demonstrate the validity and utility of our metric for LLM evaluation.We will revise the manuscript to clarify this and ensure accurate representation of our findings. Thank you for your valuable feedback.
>
> ---
>
> **Q6：The statements in Figure 4 is inaccurate**
>
>  Thank you for pointing this out. Upon review, we acknowledge that the claim regarding convergence in Figure 4 was inaccurate. In the revised manuscript, we have removed any mention of convergence and updated the discussion to focus solely on the observed trends.  We appreciate your feedback, which helped us ensure the accuracy and clarity of our findings.

---

> ### Author Response · Authors · 2024-11-22
> **(3/6)**
>
> **Q7 & Q8 & Q10：**
>
> - The proofs lack formal rigor, often omitting key arguments needed to substantiate the results.
>
> - Notations are introduced without clear definitions.
> - The proof of Theorem 1 is not complete: some (main) arguments are not exposed.
>
> Thank you for bringing this to our attention. We will thoroughly review and clearly define all notations in the revised manuscript to ensure consistency and clarity. To address this concern, we have provided a more detailed and rigorous proof in the revised manuscript, explicitly including all key arguments necessary to substantiate the results. We are confident that these updates will enhance the precision and readability of our work.
>
> \section{Proof of Theorem 1}
>
> \label{sec:Theorem1}
>
> This section presents the proof of the strictly inverse monotonic relationship between the entropy \( H(A) \) and the Frobenius norm \( $\|A\|_F$ \) for a matrix $A$ .
>
> \subsection{Definitions and Assumptions}
>
> Let \( $A \in \mathbb{R}^{B \times C}$ \) be a non-negative matrix where each row represents a probability distribution:
>
> $$
> \sum _ {j=1}^C A _ {i,j} = 1, \quad \forall i = 1, 2, \ldots, B
> $$
>
> with \( $A _ {i,j} \geq 0$ \). Here, \( $A _ {i,j}$ \) denotes the predicted probability that sample \( i \) belongs to category \( j \).
>
>
>
> The **Shannon entropy** $H(A)$  of the matrix  $A$  is defined as:
>
> $$
> H(A) = -\frac{1}{B} \sum_{i=1}^B \sum _ {j=1}^C A _ {i,j} \log(A _ {i,j})
> $$
> where \( $0 \log(0)$ \) is defined as \( 0 \) by convention.
>
>
>
> The **Frobenius norm** \( $\|A\|_F$ \) is defined as:
>
> $$
> \|A\| _ F = \sqrt{\sum _ {i=1}^B \sum _ {j=1}^C A _ {i,j}^2}.
> $$
>
>
> \subsection{Step 1: Entropy and Frobenius Norm for a Single Row}
>
> Consider a single row \( $\mathbf{a} = [a_1, a_2, \ldots, a_C]$ \), where \( $a_j = A_{i,j}$ \), \( $a_j \geq 0$ \), and \( $\sum_{j=1}^C a_j = 1$ \). The **row entropy** is:
>
> $$
> H_i = -\sum_{j=1}^C a_j \log(a_j),
> $$
> and the **row Frobenius norm** is:
>
> $$
> \|\mathbf{a}\|  _2 = \sqrt{\sum _ {j=1}^C a _ j^2}.
> $$
>
>
> To determine the extrema of \( H_i \), we use the method of Lagrange multipliers. Define the Lagrangian:
>
> $$
> L(a_1, a_2, \ldots, a_C, \lambda) = -\sum_{j=1}^C a_j \log(a_j) + \lambda \left( \sum_{j=1}^C a_j - 1 \right).
> $$
>
>
> Taking the partial derivatives and setting them to zero:
>
> $$
> \frac{\partial L}{\partial a_j} = -\log(a_j) - 1 + \lambda = 0 \implies a_j = e^{\lambda - 1}.
> $$
> Using the normalization condition \( $\sum_{j=1}^C a_j = 1$ \), we find \( $a_j = \frac{1}{C}$ \) for all \( $j$ \). Substituting \( $a_j = \frac{1}{C}$ \) into \( $H_i$ \) and \( $\|\mathbf{a}\|_2$ \):
>
> $$
> H_i = \log(C), \quad \|\mathbf{a}\|_2 = \sqrt{\frac{1}{C}}.
> $$
>
>
> For the minimum entropy, let \( $a_k = 1$ \) and \( $a_j = 0$ \) for \( $j \neq k$ \):
>
> $$
> H_i = 0, \quad \|\mathbf{a}\|_2 = 1.
> $$
>
>
> Thus, \( $H_i$ \) and \( $\|\mathbf{a}\|_2$ \) exhibit an inverse monotonic relationship.
>
>
>
> \subsection{Step 2: Generalizing to the Entire Matrix}
>
> The matrix-level entropy \( $H(A)$ \) and Frobenius norm \( $\|A\|_F$ \) are given by:
>
> $$
> H(A) = \frac{1}{B} \sum _ {i=1}^B H _ i, \quad \|A\| _ F = \sqrt{\sum _ {i=1}^B \|\mathbf{a} _ i\| _ 2^2}
> $$
>
>
>
>
> Since  $H(A)$  is the average of row entropies and \( $\|A\|_F$ \) is derived from the sum of row Frobenius norms, the inverse monotonicity established for a single row generalizes to the entire matrix.
>
>
>
> \subsection{Step 3: Bounds for \( $\|A\|_F$ \)}
>
> To determine the bounds for \( $\|A\|_F$ \):
>
> - **Maximum Frobenius Norm**: When each row is a one-hot vector (minimum entropy):
>
> $$
> \|A\| _ F = \sqrt{\sum _ {i=1}^B 1} = \sqrt{B}
> $$
>
> - **Minimum Frobenius Norm**: When each row is uniformly distributed (maximum entropy):
>
> $$
> \|A\| _ F = \sqrt{\sum _ {i=1}^B \sum _ {j=1}^C \left(\frac{1}{C}\right)^2} = \sqrt{\frac{B}{C}}
> $$
>
>
> \subsection{Step 4: Implications for Model Evaluation}
>
> The inverse monotonic relationship between \( $H(A)$ \) and \( $\|A\|_F$ \) implies that models with higher \( $\|A\|_F$ \) exhibit greater discriminability and certainty in their predictions. This makes \( $\|A\|_F$ \) a useful proxy for evaluating the compression and confidence capabilities of large language models (LLMs).
>
>
>
> \subsection{Conclusion}
>
> The proof establishes that \( $H(A)$ \) and \( $\|A\|_F$ \) are strictly inversely monotonic. This relationship provides theoretical justification for using \( $\|A\|_F$ \) as an evaluation metric in LLMs, where balancing diversity and confidence is essential.
>
> ---
>
> **Q9：explicitly defining the objective of measuring the nuclear norm (or rank) of the output matrix**
>
> Thank you for pointing this out. The nuclear norm(or rank) is used to evaluate the compression capabilities of LLMs by providing a convex approximation of matrix rank. This reflects the effective dimensionality of embeddings and helps quantify the balance between information richness and compactness. We will clarify this objective in the revised manuscript for better understanding.

---

> ### Author Response · Authors · 2024-11-22
> **(4/6)**
>
> **Q11：$C_p(A)$ definition?**
>
> Thank you for your feedback regarding the need to clarify $C_p(A)$ in Equation (5). In the revised manuscript, we provide a formal definition of $C_p(A)$ that better aligns with the continuous and high-dimensional nature of LLM generation tasks.
>
> We now define  $C_p(A)$  as follows, with the final version of our paper incorporating this definition immediately **in Equation (5)**.
>
> $$
> C_p(A) = \text{rank}\left(\prod [A_{i, \text{arg max}(A_i)}]\right),
> $$
>
> where $C_p(A)$ reflects the effective number of active dimensions in the output matrix $A$. In LLM generation tasks, $A$ represents the model's continuous activation matrix across embedding dimensions for a given batch of inputs. Unlike discrete categories in classification tasks, here $C_p(A)$ captures the diversity and richness of activations by evaluating the rank of a transformed matrix. The transformation uses an $\text{argmax}$ operation to identify the most significant activations for each input, linking $C_p(A)$ to the extent of dimensional utilization in the representation space.
>
> To further ensure clarity, we revised the surrounding text to explain how $C_p(A)$ integrates into Equation (5). This revision highlights how $C_p(A)$ quantitatively measures the diversity of active dimensions in $A$, providing insights into the efficiency and comprehensiveness of the model's embedding space utilization during text generation.
>
> We sincerely appreciate your comment, which allowed us to refine this section and improve the theoretical rigor of the manuscript.
>
> ---
>
> **Q12：the definition of top() in Eq.9**
>
> Thank you for pointing out the need for greater clarity regarding the "top" notation in Eq. (9). In the revised manuscript, we have added an explicit definition of "top" immediately before Eq. (9). Specifically, "top" refers to selecting the \( D \) largest values from the given set. This operation is central to our approach, as it enables the approximation of the nuclear norm using the  $L_{1,2}$ -norm.
>
> Additionally, we have revised the surrounding text to better illustrate how this operation integrates with the overall computational framework and impacts the efficiency of our method. We appreciate your feedback, which has helped us improve the precision and clarity of the manuscript.
>
> ---
>
> **Q13：Notations and indices are not consistent throughout the paper.**
>
> We appreciate your valuable observation regarding the consistency of notations and subscripts. To address this, we have conducted a thorough review of the manuscript to ensure that all notations and subscripts are used consistently throughout. This includes standardizing definitions across equations, algorithms, and figures to enhance readability and coherence. Thank you for highlighting this issue, which has significantly helped us refine the presentation of our work.
>
> ---
>
> **Q14：$\Sigma _ {S _ i} $ in Eq.(13) is never properly defined.**
>
> Thank you for pointing out the need to clarify the definition of \( $\Sigma _ {\mathcal{S} _ i}$ \). In the revised manuscript, we have explicitly defined \( $\Sigma _ {\mathcal{S} _ i}$ \) in the section where the formula is introduced. Specifically, we now state:
>
>
> $$\Sigma _ {\mathcal{S} _ i} = \sum _ {j=1}^d \mathcal{S} _ {i,j}$$
>
> where \( $\Sigma_{\mathcal{S}_i}$ \) represents the sum of all elements in the embedding vector \( $\mathcal{S}_i$ \), and  $d$  denotes the embedding dimension. This definition ensures that the matrix entropy calculation is normalized across the dataset.  The updated definition has been incorporated into the main text **after Eq. (13)** to ensure self-containment and clarity. Thank you for your suggestion, which has significantly enhanced the precision and comprehensibility of our manuscript.

---

> ### Author Response · Authors · 2024-11-22
> **(5/6)**
>
> **Q15 & Q16 & Q22：**
>
> - S 5.2.1 the authors conclude in l.341 that the metric considered achieves comparable evaluation effectiveness to Matrix Entropy, however, the figure presented for this experiment only compares time complexities.
>
> - In l.379, the authors claim that their metric showcases superior accuracy, however, the meaning of this sentence is not clear to me. In which sense did the author show that the metric considered is more effective than others?
> - How one can compare the proposed metric with others? More precisely, in which sense the proposed metric demonstrates superior accuracy compared to traditional metrics?
>
> Thank you for pointing this out. We recognize that our wording may have been unclear, and we apologize for any confusion it caused. Metrics are not inherently superior or inferior; rather, they are designed to serve specific purposes based on the evaluation context.
>
> What we intended to convey is that our proposed metric produces evaluation trends consistent with those observed in validated metrics such as perplexity and Matrix Entropy. Specifically, our results demonstrate that as model size increases, performance improves, and as size decreases, performance declines—consistent with established evaluation principles for large language models. These consistent trends validate the reliability and utility of our metric as an effective tool for LLM evaluation.
>
> Additionally, our metric offers significant practical advantages by reducing computational complexity and evaluation time, which enhances its feasibility for large-scale model assessments. We will revise the manuscript to clearly articulate these points and ensure a precise representation of our findings. Thank you for your constructive feedback, which has helped us refine our presentation.
>
> ---
>
> **Q17：typo l. 115: Shannon Entropy only applies to nonnegative matrices**
>
> Thank you for pointing this out. We acknowledge the importance of clearly specifying such details. The statement that "Shannon Entropy only applies to nonnegative matrices" has already been addressed in the supplementary materials ( **Please line 1046, A5 in appendix** ), where we provided a detailed proof to support this. We will ensure that this clarification is also reflected in the main text to avoid any ambiguity.
>
> ---
>
> **Q18：What is the purpose of evaluating the nuclear norm of the output matrix?**
>
> Thank you for your question. The nuclear norm of the output matrix is evaluated to assess the model's ability to extract meaningful information while minimizing redundancy. As a convex approximation of matrix rank, the nuclear norm captures the effective dimensionality of the embeddings, providing insight into the model’s ability to generate structured and diverse representations. This metric effectively measures the compression capabilities of large language models (LLMs), reflecting how efficiently the model retains essential information while reducing redundancy. We believe this property makes the nuclear norm a valuable tool for evaluating LLM performance.
>
> ---
>
> **Q19：If the goal is to measure the diversity of the prediction, why not consider directly the entropy as defined in Eq. (1)?**
>
> Thank you for the question. The primary goal of our metric is not merely to measure prediction diversity but to evaluate the model's information compression capability—its ability to extract meaningful information from large datasets while minimizing redundancy. Unlike entropy or loss, which focus solely on the final output, our metric evaluates the information captured in the hidden layers, providing a deeper assessment of the model's representation and compression efficiency. This makes it particularly suitable for evaluating LLMs.
>
> ---
>
> **Q20：In which sense does the nuclear norm enable the measurement of the compression capabilities of LLMs?**
>
> Thank you for the question. The nuclear norm enables the measurement of the compression capabilities of LLMs by serving as a convex approximation of matrix rank. Matrix rank reflects the amount of independent information captured in a representation, which directly relates to a model's ability to compress data by retaining essential features while reducing redundancy. Additionally, the nuclear norm is an upper bound of the Frobenius norm, linking it to the energy preserved in the hidden states. By minimizing the nuclear norm, we effectively assess the model's ability to achieve low-rank approximations, which is a key aspect of information compression in high-dimensional representations. This property makes it particularly relevant for evaluating LLMs' efficiency in extracting structured information.
>
> ---

---

> > ### Author Response · Authors · 2024-11-22
> > **(6/6)**
> >
> > **Q21：Is the proposed estimator an efficient approximation of the nuclear norm in practice?**
> >
> > Yes, our method uses the $L_{1,2}$  norm to approximate the nuclear norm efficiently and effectively. While nuclear norm calculation via SVD has a complexity of  $O(n^3)$ , the  $L_{1,2}$  norm reduces this to  $O(n^2)$ , making it suitable for large-scale models.

---

> > > ### Comment · Reviewer_6Meo · 2024-11-30
> > >
> > > Thank you for your very detailed reply. I think the authors have made a great effort to improve their work and have answered most of my concerns, which is why I've decided to increase my score. However, I think there are also a lot of changes to implement, and so I think that currently, the work is still below the acceptance bar.

---

> > > > ### Author Response · Authors · 2024-11-30
> > > > **Response to Reviewer 6Meo**
> > > >
> > > > Dear Reviewer,
> > > >
> > > >
> > > > We sincerely appreciate your positive feedback and for raising your rating of our paper. We're glad our revisions met your expectations, and your constructive comments were invaluable in helping us improve our work.
> > > >
> > > >
> > > > Best regards

---

### Official Review · Reviewer_W8Xn · 2024-11-03

**Soundness:** 2
**Presentation:** 2
**Contribution:** 2
**Rating:** 5
**Confidence:** 3

**Summary:**

The paper proposes using matrix nuclear-norm as a new metric to evaluate the LLM performance. The major benefit is the computational efficiency can reduce to O(n^2) as compared with the recently new metric matrix entropy. A set of experiments were conduced to show the matrix nuclear-norm evaluation/ranking is consistent with other metrics.

**Strengths:**

1. proposing a new metric for LLM evaluation. As the hottest area in ML/AI, contributing a new metric for LLM evaluation might be fundamental and impactful.

**Weaknesses:**

1. The biggest concern from my end is that the contribution might be incremental and limited. The work is motivated by matrix entropy, a very recently proposed metric (a few month ago), which has not fully established the value in LLM community. I'd be quite conservative to its true usage or value before considering any efficiency improvement on it.

2. The efficiency improvement from O(n^3) to O(n^2) is not valuable in practice. If n is small (e.g., less than 1K), such complexity change can be ignored. On the other hand, if n is super large, say, 10K or 100K, O(n^2) is far from being adequate to be scalable. It will be good to provide some algorithms or approximations that attains O(n) or O(n log n) time complexity that is "truly" efficient and scalable.

3. The presentation and notation are quite confusing and/or contradiction, yet without clarification. For example, in Eq(1), i believe elements of A are real-valued probabilities, then in line 143, it claims "involves the number of unique categories in matrix A". What does it mean? Isn't the number of unique categories the column number of A or something else? in Line 188, it says "since A is typically sparse, with only a few non-zero responses in each category". Following eq(1), A should be a dense probability matrix, then why is A typically sparse? is A now a binary matrix?

**Questions:**

see above

---

> ### Author Response · Authors · 2024-11-22
> **(1/2)**
>
> **Q1: The concern on novelty, the contributions may be limited and incremental.**
>
> Thank you for your thoughtful feedback. We understand your concern regarding the novelty and incremental nature of the contributions. It is worth noting that the metric we build upon, originally termed Matrix Entropy, has been formally renamed as *"Diff-eRank: A Novel Rank-Based Metric for Evaluating Large Language Models"* and has been accepted at NeurIPS 2024 after rigorous peer review. This acceptance underscores the metric’s growing recognition and relevance within the LLM community. Our work builds on this foundation by addressing its computational inefficiency, a critical limitation in its practical application to large-scale models. Specifically, we propose an efficient approximation method that significantly reduces the computational complexity, making the metric scalable and applicable to real-world scenarios. This improvement enhances the practicality and impact of Diff-eRank, ensuring its utility in evaluating modern LLMs.
>
> **Q2: The efficiency improvement from O($n^3$)  to O($n^2$) is not valuable in practice. Actually, the fundamental research on optimizing computational efficiency can not be that astonishing, it should be conducted step-by-step and with some mathematical support.**
>
> Thank you for your insightful feedback regarding the practical value of the efficiency improvement from \( $O(n^3)$ \) to \( $O(n^2)$ \). While we understand your perspective, we believe that computational efficiency optimization, as a foundational area of research, is a step-by-step process requiring rigorous mathematical support and theoretical justification. Such advancements are rarely achieved instantaneously or without careful derivations, and our work aims to contribute to this ongoing effort with a solid theoretical foundation and empirical validation.
>
> In real-world applications, where large language models (LLMs) often have massive parameter sizes, metrics that can efficiently evaluate their performance are particularly critical. In Section 5.2.1 (l.317), we provide experimental evidence demonstrating the practical significance of our approach. For instance, when applied to models of increasing scale, such as a 13B-parameter model, our method completes the computation in 0.82 hours, compared to 16 hours required by Matrix Entropy. This significant reduction in computational time highlights the practical relevance of our metric, particularly as model sizes continue to grow.
>
> While achieving \( $O(n)$ \) or \( $O(n \log n)$ \) complexity is an aspirational goal, our proposed method represents a meaningful step forward, making evaluation feasible for large-scale models and contributing to the broader understanding of computational efficiency in the context of LLM evaluation.

---

> ### Author Response · Authors · 2024-11-22
> **(2/2)**
>
> **Q3: The presentation is poor and some notations are confusing.**
>
> Thank you for pointing this out. We acknowledge the need for greater clarity and consistency in our notations and presentation. Below, we clarify the key points and outline our planned revisions:
>
> - **Matrix $A$ as an Output Matrix:**
>   In the context of evaluating LLMs, the matrix $A$ represents the hidden layer outputs or intermediate embeddings generated by the model for a given batch of inputs. Each element $A_{i,j}$ corresponds to the activation value of the $j$-th dimension for the $i$-th input. Unlike a probability matrix typically used in classification tasks, $A$ serves as a high-dimensional representation that captures the model's internal computations. The values and structure of $A$ provide insights into the diversity and richness of the model's output, making it a crucial component for evaluating efficiency, representation capacity, and redundancy within LLMs.
>
> - **Definition of "Active Dimensions" in $E_C$:**
>   The term "active dimensions" refers to the subset of dimensions in the output matrix $A$ that contribute significantly to the model's representations, determined by their activation values. These dimensions are typically identified using a thresholding criterion (e.g., non-zero values or the top-$k$ largest activations). Unlike the total number of columns in $A$, which represents all potential dimensions, $E_C$ measures the average number of active dimensions across the dataset. This serves as an indicator of the model's ability to effectively utilize its representation space, balancing compactness with information richness.
>
> - **Sparsity of $A$:**
>   While $A$ is formally dense, in practice, many elements are near zero, especially for high-dimensional models or tasks with complex representation spaces. This sparsity arises naturally in LLMs, as meaningful information tends to be concentrated in a small number of dimensions, leaving many activations negligible. Such sparsity reflects the model’s efficiency in encoding relevant features and supports computational optimizations like nuclear norm approximation. By leveraging this sparsity, our approach reduces computational overhead while maintaining evaluation effectiveness.
>
> - **Clarifying $A$ is Not Binary:**
>   Despite its sparsity, $A$ remains a real-valued matrix with continuous activation values. The sparsity does not imply binary (0/1) values but rather a selective distribution of activations where most dimensions contribute minimally. This allows the matrix to encode nuanced information in the most critical dimensions while discarding less relevant features, a key characteristic for evaluating the compression and efficiency of LLM representations.
>
> To address these issues, we will ensure in the revised manuscript:
> 1. All key notations, including  $A$ ,  $E_C$ , and their properties, are clearly defined when first introduced.
> 2. Logical consistency in the use of terms like "dense," "sparse," and "unique categories."
> 3. Improved explanations of  $E_C$  and sparsity in the context of model evaluation metrics.
>
> We appreciate your feedback, which has guided us to improve the clarity and logical coherence of our work.

---

> > ### Author Response · Authors · 2024-12-02
> > **Eagerly anticipating your insights and feedback.**
> >
> > Dear Reviewers,
> >
> > We sincerely appreciate the time and effort you have dedicated to reviewing our manuscript and for the valuable suggestions you have provided.
> >
> > As there will be no second round of author-reviewer discussions, we are in urgent need of your recommendation by December 3rd. With the author-reviewer discussion phase coming to a close, we would like to confirm if our responses have adequately addressed your concerns.
> >
> > We have provided detailed responses to your concerns a few days ago, and we hope these have sufficiently addressed the issues you raised. Should you require further clarification or have additional concerns, please feel free to contact us. We are more than willing to continue the dialogue with you.
> >
> > Best regards,
> >
> > Authors

---

### Author Response · Authors · 2024-11-28
**Kind Reminder**

Dear Reviewers,


We sincerely appreciate your thoughtful and constructive feedback during the review process. Your insights have been invaluable in strengthening our manuscript. In response to your comments, we have made the necessary revisions, and all changes have been highlighted in blue for your convenience.

We believe that the revisions address your concerns, and we would be happy to provide any further clarifications or engage in further discussion if needed.

Thank you once again for your time, expertise, and consideration.


Best regards

---

### Meta-Review · Area_Chair_2WXe · 2024-12-19

**Metareview:**

Dear Authors,

Thank you for your valuable contribution to ICLR and the ML community. Your submitted paper has undergone a rigorous review process, and I have carefully read and considered the feedback provided by the reviewers.

This work proposes proposes a measure of the compression abilities of large language models (LLMs) using a modified norm called "matrix nuclear norm" that approximates the nuclear norm of the output probabilities obtained from LLMs, hence avoiding singular value decomposition.

The paper received borderline final review scores (5,5,5,6). Reviewers pointed out certain critical issues including (i) lack of motivation and justification of using nuclear norm and matrix-nuclear-norm for evaluation of LLMs (considering matrix entropy is not a well-established evaluation metric for LLMs), (ii) limited novelty of the contribution, (iii) lack of clarity of the presentation and notational issues, (iv) issues with the proof of the theoretical result. The authors provided a detailed rebuttal, however, it was not convincing enough for the reviewers to increase their scores.

Given the current form of the paper and the reviewer discussion, I regret to inform you that I am unable to recommend the acceptance of the paper for publication at ICLR. I want to emphasize that this decision should not be viewed as a discouragement. In fact, the reviewers and I believe that your work has valuable insights and, with further development and refinement, can make a meaningful impact on the field.

I encourage you to carefully address the feedback provided by the reviewers and consider resubmitting the paper. Please use the comments and suggestions in the reviews to improve and refine your work.

Best,
AC

**Additional Comments On Reviewer Discussion:**

Many reviewers pointed out a series of critical issues including (i) lack of motivation and justification of using nuclear norm and matrix-nuclear-norm for evaluation of LLMs (considering matrix entropy is not a well-established evaluation metric for LLMs), (ii) limited novelty of the contribution, (iii) lack of clarity of the presentation and notational issues, (iv) issues with the proof of the theoretical result. The authors provided a detailed rebuttal, however, it was not convincing enough for the reviewers to increase their scores.

---

### Decision · Program_Chairs · 2025-01-22

Reject